# THE GEOMETRY OF ATTENTION: RICCI CURVATURE AND TRANSFORMERS TRAINING AND ROBUSTNESS

## ABSTRACT

Transformer models have revolutionized machine learning, and the theoretical underpinnings behind their success are only now starting to be explored. In this work, we analyze the performance and robustness of transformers by considering the attention mechanism as a graph operator, focusing on the geometry of attention maps viewed as weighted graphs. Specifically, we investigate the role of Ricci curvature, a metric closely tied to graph spectral properties and system robustness, in shaping the training dynamics and robustness of transformers. Our theoretical analysis establishes a link between Ricci curvature and the convergence of gradient descent on transformers, and consequently, their training and fine tuning. We also show that a higher frequency of more positive values in the Ricci curvature distribution of attention graphs, therefore more system robustness, leads to more robust transformers, highlighting the impact of curvature on the robustness of transformers. Leveraging these insights, we propose an efficient regularization method to train curvature-adjusted transformers. Supporting our theoretical findings, experiments show that our proposed attention curvature manipulation can improve the learning speed, performance, or generalizability of vision and language transformers. Additionally, our observations point to a trade-off between their performance and robustness. This work demonstrates that the geometry of the attention map provides a theoretically elegant and computationally versatile framework for analyzing and manipulating transformers training, generalization, performance, and robustness, opening new avenues for designing models using geometric concepts.

## 1 INTRODUCTION

Transformers (Vaswani et al., 2017) have become the cornerstone of modern machine learning, excelling in tasks across natural language processing (Brown et al., 2020; Devlin et al., 2018) and computer vision (Dosovitskiy et al., 2020; Liu et al., 2021b; Touvron et al., 2021). Recent studies have begun to explain the inner workings of transformers, analyzing their expressive power (Yun et al., 2020; Zaheer et al., 2020), their generalization properties (Li et al., 2023; Zhou et al., 2024), in-context learning (Von Oswald et al., 2023), learning dynamics (Abbe et al., 2024; Dovonon et al., 2024), and robustness (Bhojanapalli et al., 2021; Shao et al., 2021). A rigorous understanding of how transformers process their input remains elusive, especially through mathematical tools offering practical solutions. To address this gap, we analyze transformers by treating attention mechanisms as graph operators and exploring for the first time the *geometry* of attention maps as weighted graphs. Focusing on graph Ricci curvature (Bauer et al., 2017; Ollivier, 2009), a well-established metric closely tied to graph spectral properties and system robustness (Bauer et al., 2011; Pouryahya et al., 2017), our findings reveal its connection to the training dynamics and robustness of transformers.

Our work reveals that the Ricci curvature of the attention graph plays a crucial role in shaping the training dynamics and robustness of transformers. By analyzing the geometry of attention graphs considering their spectral properties, we establish a direct link between Ricci curvature and the convergence behavior of gradient descent in transformers. Specifically, we demonstrate that Ricci curvature, with established links to system robustness, controls a proxy linked to the convergence rate during training, implied from our Lemma 4.2. Additionally, analyzing the statistical mechanics of transformers by conceptualizing the attention mechanism as a system of interacting particles, we uncover how the Ricci curvature distribution of the attention graph influences the robustness of transformers. Visually summarized in Figure 1, this dual perspective provides a comprehensive

Figure 1: A visual summary of the main concepts and contributions. The lines between each block indicate the connection between concepts and tools that are already established in the literature (dotted black), where we provide complementary reinterpretation (dotted blue), our main contributions (solid green), or otherwise novel connections established in this paper (solid purple).

understanding of how the geometry of attention maps affects both the performance and robustness of transformer models. Informed by our theoretical and empirical findings, we then introduce a simple yet efficient training method that manipulates the curvature distribution of attention maps through a variance-adjusting regularization term. Our experiments show that increasing variance speeds up loss optimization, whereas decreasing it enhances performance at the cost of reduced generalizability. These experiments also illuminate the trade-off between performance and robustness, demonstrating that changes in the curvature can enhance one at the potential expense of the other.

**Main contributions**

- We establish a theoretical link between Ricci curvature in attention graphs and gradient descent convergence in transformers.

- Using a statistical mechanics perspective, we show how the Ricci curvature distribution in attention graphs affects transformer robustness, particularly against input perturbations.

- We introduce a training method that allows for curvature adjustment in the attention graph, influencing performance and generalizability of transformers, validated by empirical results.

- Linking attention graph geometry to key aspects of transformer behavior, this paper takes the first step toward applying geometric tools to study and improve the attention mechanism.

To recap, this paper marks a step toward elucidating the success of transformers by revealing how the geometry of attention graphs influences their behavior. Through a combination of theoretical insights, empirical investigations, and practical methods, we demonstrate how understanding and manipulating this geometry can enhance training, performance, and robustness. To our knowledge, this is the first work to connect the geometry of attention to these critical aspects of transformers, offering new avenues for designing more effective transformer models, through geometric insights and techniques.

## 2 BACKGROUND

**Transformers and attention mechanism.** The core component of transformers is the attention mechanism, which allows the model to account for dependencies between input tokens (Vaswani et al., 2017). Given an input $X = (x_1, \ldots, x_n)$, where $x_i \in \mathbb{R}^d$ is the embedding of the $i$-th token, for the value matrix $V \in \mathbb{R}^{n \times d_v}$, the attention scores are computed based on pairwise similarities between the query ($Q \in \mathbb{R}^{n \times d_k}$) and key ($K \in \mathbb{R}^{n \times d_k}$) representations,

$$\text{Attention}(Q, K, V) = \text{softmax}\left(\frac{QK^\top}{\sqrt{d_k}}\right)V. \tag{1}$$

The attention matrix $A = \mathrm{softmax}\left(\frac{QK^\top}{\sqrt{d_k}}\right)$ can be viewed as the adjacency matrix of a graph, where nodes correspond to tokens and edge weights to attention scores. Focusing on an architecture common to the most widely adopted transformer models, in this paper we consider the self-attention mechanism in a weighted fully-connected graph (full attention), where $Q = XW_Q$, $K = XW_K$, and $V = XW_V$ with learnable weight matrices $W_Q, W_K, W_V \in \mathbb{R}^{d \times d_k}$.

**Graph Ricci curvature.** Ricci curvature (Figure 2) captures the extent to which the local geometry induced by a Riemannian metric deviates from that of a Euclidean space (Bauer et al., 2017). Graph Ricci curvature is a powerful tool in deep learning (Di Giovanni et al., 2023; Liu et al., 2023; Southern et al., 2023; Topping et al., 2021). We use the Ollivier-Ricci curvature (ORC) (Ollivier, 2009), an optimal transport formulation of graph Ricci curvature, defined as

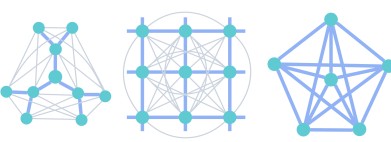

$$\kappa_{OR}(v, u) := 1 - \frac{W_1(\mu_v, \mu_u)}{d_G(v, u)}, \qquad (2)$$

for an edge $(v, u) \in \mathcal{E}$ on a graph $G = (\mathcal{V}, \mathcal{E})$, where $\mu_v$ and $\mu_u$ are probability measure on the nodes anchoring $(v, u)$, $d_G(.)$ represents a distance metric on $\mathcal{V}$, and $W_1$ denotes the 1-Wasserstein distance (Lin et al., 2011; Jost & Liu, 2014).

Figure 2: Three weighted complete graphs with highlighted larger weight edges (in blue). Different edge weight distributions result in negative (left), near zero (middle), and positive (right) Ricci curvature. Curvature in a weighted graph is negative/zero/positive when removing low-weight edges (in gray) yields a tree/grid/complete graph, respectively.

**Ricci curvature and system robustness.** Ricci curvature has been linked to system robustness (Pouryahya et al., 2017), owing to its connection with entropy. This established link is highlighted by the following fundamental result from optimal transport (Lott & Villani, 2009), which bounds entropy based on a lower bound of the Ricci curvature (Pouryahya et al., 2017),

$$S(\mu_\lambda) \geq (1 - \lambda) S(\mu_0) + \lambda S(\mu_1) + \underline{k} \frac{\lambda(1 - \lambda)}{2} W_2(\mu_0, \mu_1)^2, \qquad (3)$$

where $S(\cdot)$ denotes the Boltzmann entropy (Adkins, 1983), $\underline{k}$ bounds the Ricci curvature from below, $W_2(\mu_0, \mu_1)$ is the Wasserstein distance of order 2 between $\mu_0$ and $\mu_1$ in the metric space $(P(\mathcal{X}), W_2)$ of probability measures on $\mathcal{X}$, and $\mu_\lambda$ is a measure on the geodesic between $\mu_0$ and $\mu_1$ given by $\lambda \in [0, 1]$. This inequality suggests that Ricci curvature and entropy are positively correlated (Pouryahya et al., 2017). Furthermore, there exists a correlation between *system robustness* and entropy. System robustness refers to the ability of a system to quickly return to its stationary state following a perturbation (Demetrius & Manke, 2005).The Fluctuation Theorem (Evans et al., 1993) establishes a positive correlation between the robustness and entropy of a system, implying that system robustness and curvature are positively correlated (Pouryahya et al., 2017).

## 3 GRAPH GEOMETRY OF LEARNING REPRESENTATIONS IN TRANSFORMERS

Attention mechanisms in transformer models can be viewed as graph operators, where the tokens serve as nodes and the attention scores as edge weights in a complete graph. The geometric properties of this attention graph inherently shape the representation space, with the eigenvalues and eigenvectors dictating the direction and scale of transformations applied to each input. This, in turn, determines the geometry of the representations that the transformer learns. Our analysis demonstrates that Ricci curvature values in the attention graph are linked to the convergence rate during training as well as the robustness of transformers to input perturbations.

Informally stating our main results, in this section, we explain how viewing the attention mechanism as a graph allows us to reach these findings. We adopt two complementary perspectives, combining a mathematical one that connects our graph-based analysis to the training dynamics of transformers, and a physics-based one that interprets the transformer as a system of interacting particles. The mathematical perspective links the ORC of the attention graph to the convergence of gradient descent-based optimization, while the physics perspective explores the statistical mechanics of transformers, drawing insights from the interplay between graph geometry, dynamics, and robustness in networked

systems. Together, these viewpoints offer a comprehensive understanding of how the geometry of attention influences the transformer behavior, as we elaborate in sections 4.1 and 5 with further technical details and empirical observations.

**Geometry of attention and training transfromers.** Transformer models are typically over-parameterized (Fan et al., 2019; Liu et al., 2021a; Wang & Tu, 2020), which leads to a non-convex optimization landscape (Bassily et al., 2018; Choromanska et al., 2015; Liu et al., 2022). In such cases, fast convergence of the loss minimization relies on the Polyak-Łojasiewicz (PL) condition (Bassily et al., 2018; Polyak, 1963; Liu et al., 2022). We demonstrate that the probability of satisfying this condition is connected to the attention graph's eigenvalues, which are in turn linked to its ORC. This relationship establishes ORC as a key proxy for understanding convergence properties in transformer optimization. This is implied from Lemma 4.2, which states that a larger smallest eigenvalue in the attention map corresponds to faster convergence of gradient descent on the transformer. Meanwhile, a monotonic function of the Ricci curvature controls the minimum eigenvalue of the attention map. It follows that the ORC is linked to the convergence rate in transformers training –a link that we empirically explore in Section 5. In Section 4.1 we elaborate on the theoretical steps leading to this finding, and the full proof and assumptions are included in Appendix A. This result introduces a novel geometric perspective on loss optimization in transformers, offering new avenues for improving their training processes.

**Graph geometry, statistical mechanics, and robustness of transformers.** Extending our geometric analysis beyond training dynamics and adapting a physics perspective, we interpret the attention graph as a system of interacting particles, where tokens represent particles and attention scores quantify inter-particle interaction strengths. In particular, the output of a deep transformer could be formulated as the solution to the continuous-time approximation of attention dynamics through a forward pass, where each attention layer is modeled as a time snapshot of particle interactions. The dynamics of this particle system, detailed in Appendix B, reveal how token representations evolve through attention layers. This analogy allows us to apply principles from statistical mechanics and dynamical systems to transformers, inheriting properties of networked systems that propagate signals across interconnected units. A crucial property of such networked systems is their robustness to signal perturbation, which for transformers translates to maintaining performance on noisy inputs. This robustness is linked to the system entropy and, consequently, to Ricci curvature (Demetrius & Manke, 2005; Pouryahya et al., 2017), providing a theoretical basis for using curvature to assess and enhance transformer robustness. As the model processes inputs through a forward pass, we observe shifts in the ORC distribution, trending toward more positive values or reducing the frequency of smaller curvature values. This observation is in line with previous findings on emergent clustering in attention layers through a forward pass (Geshkovski et al., 2023) (see Appendix B), due to the connection between ORC and weight clustering, which we explain in Appendix C. We discuss the details of how the attention dynamics leads to the shifts in the ORC distribution in Appendix B, where we also show this phenomenon on additional transformer models. These shifts, visualized in Figure 3, suggest a strong connection between attention mechanism dynamics and the geometric properties that influence transformer robustness. Thus, Ricci curvature emerges as a multifaceted concept in transformer analysis: it informs training dynamics, serves as a geometric signature of robustness and performance, and offers a promising tool for model improvement. Our empirical results in Section 5 further elucidate these diverse roles and their practical implications.

## 4 RICCI CURVATURE AND TRAINING OF TRANSFORMERS

In this section, we explore in detail the above-mentioned theoretical and practical implications of Ricci curvature in the training of transformers. We first establish the mathematical foundations linking the eigenvalue spectrum of the attention graph to the optimization process in over-parameterized models, which allows us to demonstrate how Ricci curvature influences gradient descent in transformers. We then propose a practical method to manipulate curvature distributions during training, which not only enables us to empirically validate our theoretical findings but also offers an approach to train enhanced transformers, following our findings on the role of curvature in their learning dynamics and robustness.

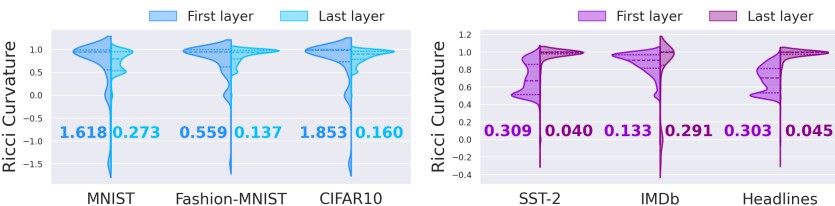

Figure 3: The ORC distributions of the first and last attention layers of ViT (blue) for a batch of MNIST, Fashion-MNIST, and CIFAR10 images, and BERT-Tiny (purple) for a batch of SST-2, IMDb, and Headlines texts. The curvature distribution shifts towards more positive values through a forward pass; quantified by the Wasserstein distances of the ORC distributions from a Dirac mass at the ORC value of 1, annotated next to each distribution (smaller means more positive leaning).

### 4.1 ATTENTION GRAPH, RICCI CURVATURE, AND GRADIENT DESCENT

In Section 3 we explained our main theoretical findings on the connection between the Ricci curvature of attention and gradient descent-based training of transformer models. Here we elaborate on the steps leading to this result, which rely on a connection we establish between the eigenvalue spectrum of transformers and the optimization of over-parameterized models. Following standard practices in the theoretical analysis of transformers (see, e.g., (Dovonon et al., 2024; Von Oswald et al., 2023; Abbe et al., 2024)), we derive our theoretical results in an idealized setting with a set of simplifying assumptions, stated here and in Appendix A. Our empirical investigation further demonstrates that the key implications of our theoretical findings extend beyond these idealized conditions, proving relevant and applicable to common transformer models in practice.

**Eigenvalues of transformers.** Consider a single-head transformer of depth $L$, with each block containing an attention layer, a skip-connection, and a projection. For this analysis, we assume the feed-forward layer in the attention block is a projection by $W_{p,l}$. Hence, the output of each layer is

$$X_{l+1} = X_l + A_l X_l W_{V,l} W_{p,l}, \tag{4}$$

where $W_{V,l}$ is the value weight matrix defined in Section 2, for block $l$. Following Dovonon et al. (2024), we further assume that all attention blocks are identical and remove the layer indexing. Stacking the column to obtain a vectorized input, $\mathbf{x}$, we can write

$$F(\mathbf{x}) = \mathbf{x}_L = (I + H \otimes A)^L \mathbf{x}, \tag{5}$$

$$\lambda^F_{ij} = \left(1 + \lambda^A_i \lambda^H_j\right)^L, \tag{6}$$

where $F(\cdot)$ and $\lambda^F_{ij}$ denote the transformer and its eigenvalues, $H := W_p^T W_V^T$, and $\lambda^A_i$ and $\lambda^H_j$ are the eigenvalues of $A$ and $H$. Note that relaxing the above-mentioned assumptions is rather straightforward, though distracting and not amenable to the here provided analysis.

**Tangent kernel and convergence of gradient descent.** Transformers are often over-parameterized (Fan et al., 2019; Liu et al., 2021a; Wang & Tu, 2020) and hence essentially non-convex (Bassily et al., 2018; Choromanska et al., 2015). The Polyak-Łojasiewicz (PL) condition (Bassily et al., 2018; Polyak, 1963) implies exponential convergence of gradient descent in non-convex regimes (Liu et al., 2022). Given a model $f_\theta(\cdot)$ parameterized by $\theta$, we say the loss function $\mathcal{L}(\theta)$ is $\mu$-PL if $\|\nabla \mathcal{L}(\theta)\|^2 \geq \mu L(\theta)$. Let $K_\theta := (\nabla f_\theta)(\nabla f_\theta)^T$ and $\lambda_1(K_\theta)$ be the tangent kernel of $f$ and its minimum eigenvalue. For any region on the loss manifold $\mathcal{S}$, the following holds:

**Theorem 4.1** (Liu et al. (2022)). $\mathcal{L}(\theta)$ is $\mu$-PL if $\lambda_1(K_\theta) \geq \mu$ for all $\theta \in \mathcal{S}$.

It follows that the learning behavior of transformers depends on the minimum eigenvalue of their tangent kernel. This suggests a connection between ORC and the training of transformers, given the link between ORC and the eigenvalue spectrum. We specify this connection next.

**Ricci curvature and convergence of gradient descent.** Given the tangent kernel of the transformer, $\varphi := (\nabla F)(\nabla F)^T$, from equations 5 and 6, we can compute the eigenvalues of $\varphi$. Thus, using the findings from (Dovonon et al., 2024) on $\lambda^A_i$ and $\lambda^H_j$ and in light of the implications of Theorem 4.1, under a set of standard assumptions (Appendix A), we prove the following lemma:

**Lemma 4.2.** *Let $p$ denote the probability of exponential convergence of gradient descent on the transformer. Then, $\frac{\partial p}{\partial \lambda^{A}_{1}} \geq 0$, i.e., the minimum attention eigenvalue is positively correlated with $p$.*

Meanwhile, monotonic functions of the minimum ORC bound the spectrum of the normalized Laplacian Bauer et al. (2011) (see Appendix A). Since the attention scores are outputs of the softmax activation, as we describe in Appendix A, there is a simple connection between the eigenvalues of $A$ and its Laplacian, which in turn implies that the spectrum of $A$ is also bounded by monotonic functions of the minimum ORC. It then follows from Lemma 4.2 that $p$ can be adjusted through shifting the distribution of ORC values of the attention matrix. This suggests that the ORC distribution bears implications for training transformers via gradient descent-based algorithms. Next, we propose a novel training heuristic to manipulate the ORC distribution in transformers, and we empirically validate this theoretical argument.

## 4.2 TRAINING CURVATURE-ADJUSTED TRANSFORMERS

Section 3 discusses the novel connections we establish between attention maps curvature and various aspects of transformer behavior. This raises an intriguing possibility: manipulating the curvature distribution of the attention graph could allow us to adjust their behavior and improve their capabilities. A well-established tool to diffuse the curvature over a manifold is the Ricci flow (Hamilton, 1988), and its discrete variation could be applied to flatten the graph and reduce the frequency of edges with highly negative curvature (representing unstable networks) (Jin et al., 2008; Ni et al., 2019). However, each iteration of the discrete Ricci flow requires computing the curvature distribution, and hence it is computationally taxing to incorporate in the training. To overcome this challenge via a computationally efficient proxy, we introduce a simple, efficient, and effective method to manipulate the ORC distribution of attention maps through regularization by a function of the variance of attention scores. Aiming to either promote or reduce the frequency of low-curvature edges, this regularization strategy not only provides a means to empirically validate our theory, but also offers a practical mechanism to enhance the training and robustness of transformers by adjusting its curvature. The experiments in this section are performed using *L-ViT* and BERT-Tiny, described in Appendix D.

**Encouraging or discouraging low-curvature edges via regularization** Recognizing the correspondence between the weight distributions and ORC in a complete graph (of full attention), we propose a regularization method to control the attention variance to adjust the ORC distribution. Negative ORC corresponds to weight imbalance, while a weight distribution with a small variance corresponds to a positively-curved graph. Hence, we expect variance increase to promote smaller ORC values, while variance decrease will lead to their elimination. In Appendix C we elaborate on how the formulation of the ORC implies its connection with weight variance. To put this in practice, we regularize the loss function with a variance-dependent term $h(\sigma^2)$, such that $\frac{\partial h}{\partial \sigma^2} < 0$ for promoting low-curvature edges and $\frac{\partial h}{\partial \sigma^2} > 0$ for reducing their frequency. The total loss is

$$\mathcal{L}_{\text{total}} = \mathcal{L}_{\text{task}} + \gamma h(\sigma^2_A), \tag{7}$$

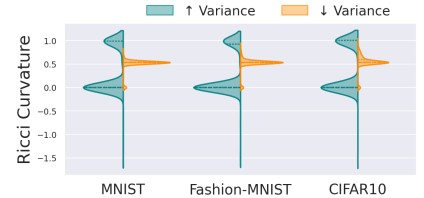

where $\mathcal{L}_{\text{task}}$ denotes the usual loss associated with the task (e.g., classification loss in our experiments), $\gamma$ the regularization coefficient, and $\sigma^2_A$ the variance of attention scores averaged over all layers. In our implementation, we use $h(x) = \exp(-ax + b)$ and $h(x) = \log(ax+b)$ to increase and decrease variance (respectively), with $a > 0$ and $b$ serving as tunable scale and shift parameters, helping to place the regularization value in an impactful range. Furthermore, the choice of exponential and logarithmic functions ensures decreasing marginals, limiting adjustments to the loss and avoiding extreme outcomes where attention either becomes uniformly distributed (losing discriminative power) or overly imbalanced (leading to instability or absurd attention).

Figure 4: Final layer ORC distributions in *L-ViT* trained via regularization methods increasing (↑) and decreasing (↓) variance, exhibiting a connection with the direction of variance regularization.

**Empirical results on curvature-adjusted training.** Our experiments on standard image and text datasets using *L-ViT* and BERT-Tiny confirm that our proposed regularization influences the ORC

distribution as anticipated, illustrated in Figure 4 for *L-ViT* and in Appendix E.1 for BERT-Tiny. Equipped with this method, we now empirically validate the theoretical findings discussed in Section 4 by comparing transformers with smaller and larger minimum values of attention ORC, corresponding to increasing and decreasing variance. Our experimental results in Figure 5 indicate that increasing variance not only accelerates loss minimization but also leads to earlier convergence. The observations on text datasets, included in Appendix E.1, echo the same findings. This supports the implications of Lemma 4.2, which point to a link between ORC and the probability of exponential convergence of gradient descent, corresponding to slower or faster convergence. This finding also suggests that our proposed regularization can enhance the training of transformer models. In subsequent sections we further empirically demonstrate the impact of this curvature adjustment on transformers, effectively influencing the learning dynamics, performance, and robustness characteristics of the model.

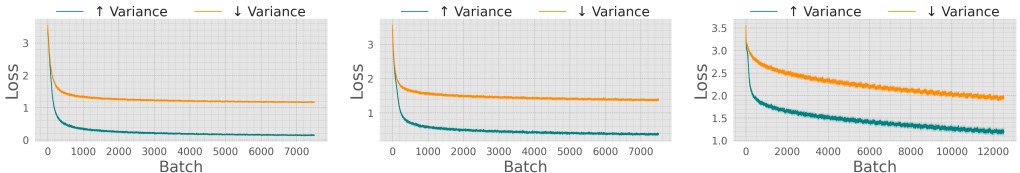

Figure 5: Training loss minimization on *L-ViT* with regularization methods increasing (↑) and decreasing (↓) variance for MNIST (left), Fashion-MNIST (middle), and CIFAR10 (right). Higher variance can promote faster loss reduction. The observations on text datasets, included in Appendix E.1, echo the same findings.

## 5 ROBUSTNESS AND CURVATURE DISTRIBUTION IN TRANSFORMERS

Building on the insights from interpreting attention as a particle system as inroduced in Section 3 and using the well-established connection between ORC and system robustness (Bauer et al., 2017; Ollivier, 2009), we now leverage the geometry of the attention graph, specifically through the ORC perspective, to explain the robustness properties of transformers. We conduct a series of experiments exploring the relationship between ORC and robustness in transformers. The discussions in sections 2 and 3 suggest that more positive ORC values, should enhance the robustness of transformers. We empirically investigate this link through two focused studies: first, by examining the curvature and robustness of pre-trained transformers to input perturbation; and second, by assessing the impact of the ORC distribution on model forgetfulness. In line with the theoretical insight, our experiments confirm that more robust transformers feature a higher frequency of larger ORC values in their attention graphs. Additional experiments in Appendix E further support these findings.

**Models and data.** The experiments in this section and Appendix E are conducted on MNIST (Deng, 2012), Fashion-MNIST (Xiao et al., 2017), and CIFAR10 (Krizhevsky et al., 2009) images for the vision models, and on the IMDb Movie Reviews (Maas et al., 2011) (IMDb), Stanford Sentiment Treebank v2 (Socher et al., 2013) (SST-2) and the News Headlines Sarcasm Detection (Misra & Arora, 2023) (Headlines) datasets for the language models. We fine-tune pre-trained vision and language transformers on these datasets, comparing ViT (Dosovitskiy et al., 2020) against DeiT (Touvron et al., 2021) for vision, and BERT-Tiny (Turc et al., 2019) against ELECTRA-Small (Clark et al., 2020) for language models. All implementation details are reported in Appendix D.

**Curvature and robustness of pre-trained transformers to input perturbation.** We now compare the performance of fine-tuned models on perturbed and unperturbed test sets (further details in Appendix D). We quantify robustness by metric $\rho$,

$$\rho := \text{accuracy}_{\text{unperturbed}} - \text{accuracy}_{\text{perturbed}}. \tag{8}$$

We find that more positive-leaning ORC distributions correspond to smaller $\rho$ values, indicating more robust models. For instance, ViT yields a $\rho$ value of 0.067 on MNIST images, which is ~20% smaller than the $\rho$ value of 0.084 obtained from DeiT. Similarly, the $\rho$ values for Fashion-MNIST and CIFAR10 datasets are ~14% and ~62% smaller in ViT than DeiT. Meanwhile, the ORC distribution in ViT is more positive-learning as quantified by the Wasserstein distance of the distribution from a $\delta$

distribution at the ORC upper bound of 1, as demonstrated in Figure 6. This pattern holds in language transformers as well, where ELECTRA-Small exhibits a more positive-leaning ORC distribution than BERT-Tiny, and is $\sim 17\%$, $\sim 49\%$, and $\sim 23\%$ more robust to input perturbation on SST-2, IMDb, and Headlines datasets. Experiments on *L-ViT* and BERT-Tiny from random initialization (Appendix E.3) show a similar trend, where training a more robust model corresponds to eliminating negative ORC values leading to a distribution concentrated around a positive mode.

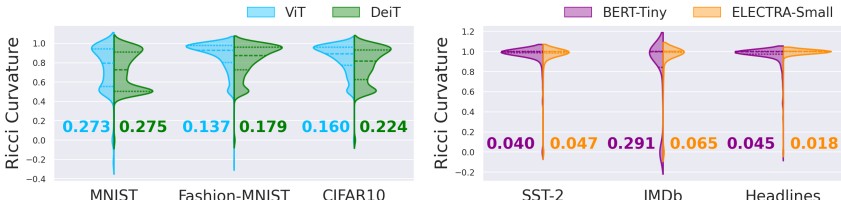

Figure 6: ORC distributions in the last attention layers of ViT and DeiT for a batch of MNIST, Fashion-MNIST, and CIFAR10 images and of BERT-Tiny and ELECTRA-Small for a batch of SST-2, IMDb, and Headlines texts. ViT (blue) and ELECTRA-Small (orange) exhibit more positive curvature values than DeiT (green) and BERT-Tiny (purple); quantified by the Wasserstein distances of the ORC distributions from a Dirac mass at the ORC upper bound of 1, annotated next to each distribution (smaller means more positive-leaning). Additional experiments in Appendix E further support these findings.

**Attention curvature and forgetfulness in pre-trained transformers.** Forgetfulness, where models lose previously learned information when exposed to new tasks or data, offers a complementary perspective on robustness, alongside robustness to input perturbation. Our experiments further link ORC in attention graphs to transformer robustness, showing that models with more positive-leaning ORC distributions are less prone to forgetfulness. We sequentially fine-tune each pre-trained model on a primary dataset and then a secondary one, tracking the performance decay on the primary dataset to measure how quickly each model "forgets" due to subsequent training. The results reveal a significant drop in classification accuracy –from above 80% to below 50%– highlighting a pronounced forgetting effect. DeiT forgets faster than ViT across all image dataset pairs (see Figure 7), and BERT-Tiny is more forgetful than ELECTRA-Small. Further details are in Appendix E.2. These findings align with our observations on perturbation robustness: ViT and ELECTRA-Small, with more positive-leaning ORC distributions and higher system robustness, show greater resistance to forgetting.

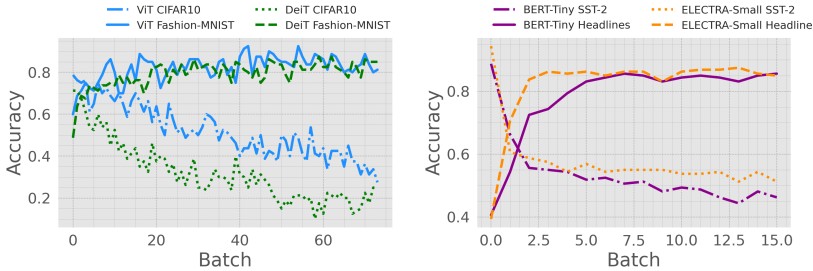

Figure 7: Test set accuracy of ViT (blue), DeiT (green) while fine-tuning on Fashion-MNIST after fine-tuning on CIFAR10 (left), and BERT-Tiny (purple) and ELECTRA-Small (orange) while fine-tuning on Headlines after fine-tuning on SST-2 (right). DeiT and BERT-Tiny demonstrate a faster degradation in performance compared to ViT and ELECTRA-Small due to subsequent fine-tuning. Further details and results are in Appendix E.

## 5.1 ROBUSTNESS AND PERFORMANCE TRADE-OFF

Our findings further highlight a potential contrast between the advantages of negative ORC for optimization and positive ORC for the robustness of transformers. The trade-off between robustness and performance is an established phenomenon studied in various contexts (Raghunathan et al., 2020; Xu & Mannor, 2006; Zhang et al., 2019). In light of the geometric underpinning of the

attention mechanism unveiled in this work, here we further explore this trade-off in the context of the distributional robustness of transformers, as indicated by their generalizability.

Table 1: Train and test accuracy and their gap using the increasing/decreasing variance regularization for six image and text datasets. Average and 95% confidence intervals over 30 trials are shown.

| | Training accuracy (↑) | | Test accuracy (↑) | | Training accuracy - Test accuracy (↓) | |
|---|---|---|---|---|---|---|
| | Decrease variance | Increase variance | Decrease variance | Increase variance | Decrease variance | Increase variance |
| CIFAR10 | **0.704 ± 0.001** | 0.613 ± 0.009 | **0.609 ± 0.002** | 0.557 ± 0.005 | 0.095 ± 0.002 | **0.056 ± 0.005** |
| Fashion-MNIST | **0.903 ± 0.000** | 0.893 ± 0.001 | **0.890 ± 0.001** | 0.883 ± 0.001 | 0.013 ± 0.001 | **0.010 ± 0.001** |
| MNIST | **0.981 ± 0.000** | 0.980 ± 0.000 | **0.984 ± 0.000** | 0.982 ± 0.000 | -0.003 ± 0.001 | -0.002 ± 0.000 |
| SST-2 | **0.986 ± 0.001** | 0.985 ± 0.001 | 0.826 ± 0.003 | **0.831 ± 0.001** | 0.160 ± 0.003 | **0.154 ± 0.002** |
| IMDb | **0.994 ± 0.000** | 0.993 ± 0.000 | 0.836 ± 0.003 | **0.838 ± 0.002** | 0.158 ± 0.003 | **0.155 ± 0.002** |
| Headlines | 0.965 ± 0.001 | **0.967 ± 0.001** | **0.915 ± 0.002** | 0.910 ± 0.002 | **0.050 ± 0.002** | 0.057 ± 0.001 |

We observe divergent impacts on performance and robustness from the opposing directions of variance regularization. Increasing the frequency of low-curvature edges by increasing variance speeds up optimization. Meanwhile, when this regularization introduces a second mode of highly positive curvature values, which is observed on all image datasets and two of the three text datasets in our experiments (Figure 4), it also improves generalizability, as evidenced by smaller train-test performance gaps. Conversely, decreasing variance enhances accuracy in most cases, while reducing generalizability. These effects are evident from Table 1. Our proposed regularization method globally impacts the attention scores and has limitations in targeted curvature adjustment, as, for instance, the simultaneous emergence of high and low curvature modes in some of our datasets suggests. Table 1 reports the results of the regularization methods, which together with the impact of each regularization on ORC distribution (figures 4 and 9), allow us to understand the impact of ORC distribution on accuracy and generalization. These results highlight the trade-off between better performance and robustness across different distributions, corresponding directly to the curvature distribution of the attention graph.

## 6 Related work

Related works are already discussed throughout the paper. In Appendix F, we provide a comprehensive literature review. Although the mentioned studies often serve as building blocks for our investigation, the link we establish in this paper between the geometry of attention maps, gradient descent-based training, performance, and robustness properties of transformers, has not been previously explored.

## 7 Conclusions, limitations, and ethical considerations

We presented an analysis of transformers through the lens of graph geometry, showing that the Ricci curvature of attention maps is a key factor in training dynamics and robustness of transformers. Our results establish a connection between the Ricci curvature of the attention graph and the convergence behavior of gradient descent in transformers. Meanwhile, we show that more positive attention curvature values correspond to more robust transformers. Our findings also indicate a trade-off between performance and robustness. Informed by these findings, we introduced an efficient curvature-adjusted regularization method that allows us to improve training or generalizability in transformers. Furthermore, we examine transformers' reliance on attention edges associated with different curvature values.

**Limitations.** A potential limitation of this work lies in the inherent difficulty of theoretical analysis of transformer models employed in practice and direct empirical validation of theoretical results derived in simplified settings. While our theory employs idealized transformers with simplifying assumptions, such simplifications are well-established traditions, due to the near intractability of fully characterizing practical transformer models. Furthermore, our empirical results are well-aligned with our theoretical discussion, confirming that our theory can extend to practical settings.

**Ethical considerations.** Our paper is theoretical and foundational in nature, but addresses the success of transformers, a family of machine learning models that has significant ethical considerations which are well-documented and are part of active research in the community.

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

# Appendix

## A  RICCI CURVATURE AND CONVERGENCE OF GRADIENT DESCENT

We establish a theoretical link between the ORC of the attention map and the convergence of gradient descent on transformers. As we discuss in Section 4.1, under a set of standard assumptions, this connection follows from the theory of non-convex optimization of over-parameterized neural networks, the eigenvalue spectrum of transformers, and the ORC bound on the graph spectrum.

Recall from Section 4.1 that given the attention matrix $A$ in a transformer $F(\cdot)$ with $L$ identical attention blocks, we can write the output of the transformer as

$$F(\mathbf{x}) = (I + H \otimes A)^L \mathbf{x}, \tag{9}$$

where $H := W_p^T W_V^T$. Our arguments implied from Lemma 4.2 state that the minimum ORC of $A$ is linked to the probability of exponential convergence of gradient descent on $F$, denoted $p$. In particular, Lemma 4.2 establishes a correlation between $p$ and the the minimum eigenvalue of $A$, and we show that this eigenvalue is bounded by a monotonic function of the minimum ORC of $A$. In this appendix, we specify the assumptions leading to this finding and provide a proof.

### A.1  ASSUMPTIONS

The majority of our assumptions follow the setup in Dovonon et al. (2024). We however introduce two other assumptions that we discuss further in this section. All assumptions are listed below.

**A0.** The transformer uses a self-attention mechanism, and the keys and queries share a projection matrix, i.e. $K = Q$.

**A1.** $A$ is positive and invertible, and all its eigenvalues are real.

**A2.** Eigenvalues of $H$ are positive and bounded, i.e., $0 < \lambda^H_1 < \ldots < \lambda^H_d < \infty$

**A3.** $\lambda^H_d < |\lambda^A_1|^{-1}$ where $\lambda^A_1$ is the smallest eigenvalue of $A$.

**A4.** Let $M := I + H \otimes A$, and let $\theta$ be the vector of the weights in $F$, consisting of the entries of the weight matrices $W_Q$, $W_K$, $W_V$, and $W_p$. Then $\nabla_\theta M$ and $M$ are simultanously diagonalizable by an orthogonal matrix $P$, i.e., there exists an orthogonal matrix $P$ and diagonal matrices $D$ and $\tilde{D}$, such that $M = P^{-1}DP$ and $\nabla_\theta M = P^{-1}\tilde{D}P$. Furthermore, we assume $\tilde{D}$ is approximately constant with respect to $D$.

**A5.** The minimum eigenvalue of $A$ is negative.

Note that assumptions **A1** and **A2** follow the setup by Dovonon et al. (2024), and imply

$$-1 < \lambda^A_1 \leq \ldots \leq \lambda^A_n = 1. \tag{10}$$

### A.2  PROOF

Considering that $F(\mathbf{x}) = (I + H \otimes A)^L \mathbf{x}$, we can write $F(\mathbf{x})$ as a linear operator $F(\mathbf{x}) = F\mathbf{x}$ where $F = (I + H \otimes A)^L$. Hence, the eigenvalues of $F$ are $\left(1 + \lambda^A_i \lambda^H_j\right)^L$.

Let $\varphi := (\nabla_\theta F)(\nabla_\theta F)^T$ be the tangent kernel of $F$. We can write $\varphi(\mathbf{x}) = \nabla_\theta M^T B B^T \nabla_\theta M$ where $M := I + H \otimes A$ and $B := M^{L-1}$. By assumption **A4**, we can write $\varphi(\mathbf{x}) = P^{-1}\tilde{D}D^{2L-2}\tilde{D}P$. Since diagonal matrices are commutative, it follows that the eigenvalues of $\varphi$ are of the form

$$\lambda^\varphi_m = \eta_m^2 \left(1 + \lambda^A_i \lambda^H_j\right)^{2L-2}, \tag{11}$$

where $\eta_m$ is a diagonal entry of $\tilde{D}$. On the other hand, it follows from **A2**, **A4**, and **A5** that the minimum eigenvalue of $\varphi$ is $\lambda^\varphi_1 = \eta_m(1 + \lambda^A_1 \lambda^H_d)^{2L-2}$ for some $m$, since by **A2** and **A4** we know $0 < 1 + \lambda^A_1 \lambda^H_j < 1$ and achieves its minimum values for $j = d$. Given $\mu$, let $p$ be the probability that $F$ is $\mu-$PL, i.e., the PL theorem holds as described in Section 4.1. From Theorem 4.1, we have

$$p = \mathbb{P}\left[\lambda^\varphi_1 \geq \mu\right]. \tag{12}$$

Note that we can equivalently write $p = \mathbb{E}\left[\mathbf{1}_{\left\{\lambda_1^\varphi > \mu\right\}}\right]$, where $\mathbf{1}_{\{\cdot\}}$ denotes the indicator function. Thus, we can write,

$$
\begin{aligned}
\frac{\partial p}{\partial \lambda^A_1} &= \frac{\partial p}{\partial \lambda_1^\varphi} \frac{\partial \lambda_1^\varphi}{\partial \lambda^A_1} \\
&= \mathbf{1}_{\left\{\lambda_1^\varphi > \mu\right\}} (2L - 2)\, \eta_1^2 \left(1 + \lambda^A_1 \lambda^H_d\right)^{2L-3} \\
&\geq 0,
\end{aligned}
$$

where the last line follows since all terms on the right-hand side of the previous line are non-negative for $L > 1$. This completes the proof of Lemma 4.2. $\quad\square$

It follows that,

$$
\Delta p \times \Delta \lambda^A_1 \geq 0, \tag{13}
$$

which establishes a positive correlation between $p$ and $\lambda^A_1$. Meanwhile Bauer et al. (2011) show that $\lambda^L_n$ is bounded above by a term that is monotonic in the minimum ORC, where $\lambda^L_n$ denotes the maximum eigenvalue of the normalized Laplacian of $A$. Notice that $A$ is row stochastic (Dovonon et al., 2024), which implies that its normalized Laplacian is in fact $I - A$. Therefore, $\lambda^L_n = 1 - \lambda^A_1$, and hence, the bound on the spectrum of the normalized Laplacian, which depends on a monotonic function of the minimum ORC, also bounds $1 - \lambda^A_1$ from above. Considering Equation 13, it follows immediately that a monotonic function of the minimum ORC controls a variable that is positively correlated with $p$, establishing the link between ORC and convergence of gradient descent. This suggests that modifying the ORC distribution of the attention map impacts transformers training.

**Discussion on the assumptions.** The self-attention mechanism is in fact a common setup for transformers, hence the only constraint assumption **A0** imposes beyond common practice is sharing the projection matrix between keys and queries. This yields a symmetric attention, while restricting the keys and queries projections to the column space of a shared weight matrix. Assumptions **A3** and **A5** are purely technical assumptions for our proof. Regarding Assumption **A4**, first note that the order in which the weight matrices are combined to parameterize $F$ with $\theta$ can be completely flexible. Therefore, it suffices if assumption **A4** holds for any arrangement of the weights in $\theta$. Despite this flexibility, being simultaneously diagonalizable restricts the degrees of freedom in $M$ and $\nabla_\theta M$. This assumption implies that the two matrices share an eigenstructure, though they could have different eigenvalues. From a geometric perspective, this assumption limits the degrees of freedom that the changes in $M$ have at each gradient step to the directions of its eigenvectors. While this is a restrictive assumption, we should keep in mind that transformers are often over-parameterized with a high degree of freedom, and hence the assumption is still plausible. Furthermore, we assume **A4** for technical reasons in order to simplify the proof of the theorem, and otherwise, the assumption is not a conceptual necessity. In other words, while assuming **A4** facilitates our proof, it is only a tool to simplify a step of the proof, not the only tool to show the main conclusion, and the implications may still hold without it.

## B  GRAPH GEOMETRY AND STATISTICAL MECHANICS OF TRANSFORMERS

In Section 3 we discussed the value of looking at transformers from the angle of networks/graphs and its consequences for training and robustness of transformers. The mathematical implications of this perspective for training transformers are provided in Section 4. Continuing in this direction from the perspective of network dynamics, in this appendix we elaborate on the physics that follow as an input signal is processed by a series of attention blocks (graph operators), which enables us to analyze the robustness properties of transformers from a networked system point of view. As we mentioned in Section 3, we interpret attention as a network of interacting particles and study the statistical mechanics of transformers in light of established results on the interplay between graph geometry and the dynamics and robustness of networked systems.

**Attention as a particle system and transformer dynamics.** A forward pass through attention blocks in transformers can be interpreted as dynamics of a system of interacting particles where each attention score represents the strength of the interaction between a pair of particles (tokens).

Leveraging the mathematical framework of Geshkovski et al. (2023), in the continuous approximation of a forward pass through deep residual network layers as evolution through time (Chen et al., 2018), the dynamics of the representations for token $i$, $z_i$ can be modeled as

$$\dot{z}_i(t) = \mathbf{P}_{z_i(t)} \left[ \frac{1}{Z_{\beta,i}(t)} \sum_{j=1}^n \exp\left\{ \beta \left[ W_Q(t) z_i(t) \right]^T W_K(t) z_j(t) \right\} W_V(t) z_j(t) \right], \qquad (14)$$

where, $Z_{\beta,i}(t)$ is the partition function, $\beta$ the system's *temperature*, and for $d$-dimensional $z$ and $u$, $\mathbf{P}_z u = u - z^T u z$ is the projection of $u$ onto the tangent space of a $(d-1)$-dimensional sphere in $R^d$ at $z$ (Geshkovski et al., 2023). The similarity between particle systems and attention mechanisms is evident from comparing equations 1 and 14. Thus, the output of the simplified transformer studied here can be approximated as the solution of Equation 14 as $t \to +\infty$. This framework sets the stage for our empirical investigation (Section 5) into how ORC distribution informs the behavior of transformers and their robustness properties in light of the discussion in Section 2.

**Forward pass and attention clustering.** Geshkovski et al. (2023) demonstrate that when the query, key, and value projection matrices are the identity, the dynamics modeled by this equation are similar to the Krause model (Krause et al., 2000), which shows particles clustering when $t \to +\infty$ (Jabin & Motsch, 2014). Clustering behavior is also observed in other similar models (Geshkovski et al., 2023; Motsch & Tadmor, 2014). Despite the differences between transformer models used in practice and the idealized model commonly assumed for analysis, the interpretation above leads to questions on attention clustering in transformers. Geshkovski et al. (2023) show empirical evidence that in commonly used pre-trained transformers the attention map becomes more clustered through a forward pass. Given the connection between ORC and clustering in networks (Jost & Liu, 2014; Sia et al., 2019), which we expand upon in Appendix C, the forward pass dynamics of attention also corresponds to an evolving pattern of ORC distribution.

**Attention dynamics and Ricci curvature.** Computing the ORC distribution in pre-trained and fine-tuned vision and language transformers, our experiments confirm our theoretically-anticipated connection between attention ORC distributions and forward pass dynamics in transformers. Empirical observations reveal that a forward pass in transformers leads to a shift in the ORC distribution towards more positive values, or significantly reduces the frequency of smaller ORC values. This is visualized for two vision and two language transformers with six standard datasets in Figure 8, and quantified by the Wasserstein distance of the distribution from a Dirac mass at 1 – maximum ORC value. A smaller distance indicates a more positive-leaning distribution in the final attention layer compared to the first layer, as observed in most cases. In the only exceptions where this Wasserstein distance value is greater – DeiT model on CIFAR10 and BERT-Tiny on IMDb – we still observe that the forward pass eliminates the smaller peak in the ORC distribution, shifting the smaller values upwards. As we explain in Section 2, ORC is a known indicator of system robustness. Meanwhile, the forward pass in transformers is known to play an important role in their remarkable performance, for instance, in their in-context learning (Von Oswald et al., 2023). Hence, these evolving patterns of ORC which follow transformers' dynamics reveal evidence on the geometric signature of both the robustness and performance of transformers.

## C  GRAPH RICCI CURVATURE AND WEIGHT DISTRIBUTIONS

The Ollivier-Ricci curvature for any given edge in the graph depends on the geodesic distances of nodes in the neighborhood of the edge. This is evident from the definition of ORC in Equation 2. To better explain this point, consider an edge $(v, u) \in \mathcal{E}$ in a graph $G = (\mathcal{V}, \mathcal{E})$, and let us expand the Wasserstein distance in Equation 2 to write the ORC in the form

$$\kappa_{OR}(v, u) \coloneqq 1 - \frac{1}{d_G(v, u)} \inf_{\gamma \in \Gamma(\mu_v, \mu_u)} \int_{\mathcal{N}_v \times \mathcal{N}_u} d_G(v', u') \, d\gamma(v', u'), \qquad (15)$$

where $d_G(\cdot, \cdot)$ is the geodesic distance over $G$, $\mathcal{N}_v$ and $\mathcal{N}_u$ denote the neighborhoods of nodes $v$ and $u$, respectively, $\mu_v$ and $\mu_u$ are probability measures over $\mathcal{N}_v$ and $\mathcal{N}_u$ and $\Gamma(\mu_v, \mu_u)$ is the set of all couplings over them.

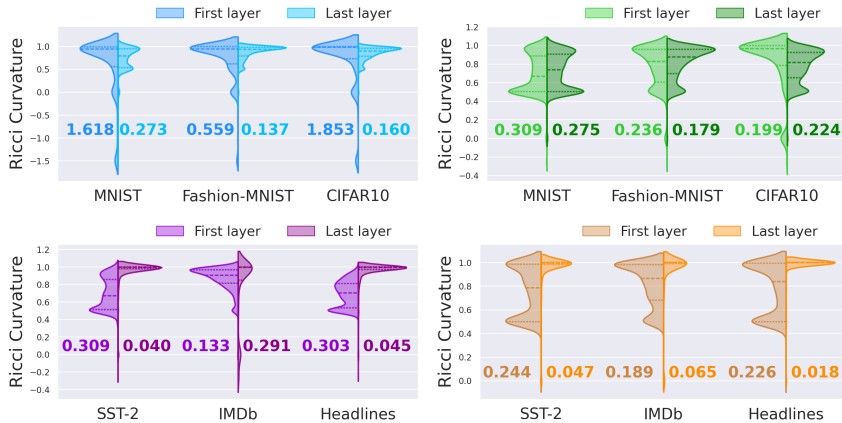

Figure 8: The ORC distributions of the first and last attention layers of ViT (blue) and DeiT (green) for a batch of MNIST, Fashion-MNIST, and CIFAR10 images, and of BERT-Tiny (purple) and ELECTRA-Small (orange) for a batch of SST-2, IMDb, and Headlines data. The curvature distribution shifts towards more positive values through a forward pass; quantified by the Wasserstein distances of the ORC distributions from a Dirac mass at the ORC value of 1, annotated next to each distribution (smaller means more positive-leaning).

**Curvature and clustering.** Note that for any pair of nodes $w$ and $w'$, $d_G(w, w')$ is the weighted shortest path between $w$ and $w'$. Hence, given two edges $(v, u)$ and $(v', u')$, the similarity of weight distributions over the neighborhoods of $(v, u)$ and $(v', u')$ leads to the similarity of geodesics in their neighborhoods, and considering the definition in Equation 15, this results in the proximity of the values of $\kappa_{OR}(v, u)$ and $\kappa_{OR}(v', u')$. Geshkovski et al. (2023) find that in a forward pass, attention maps exhibit clustering signaled by the cosine similarity of pairs of nodes, which corresponds to the similarity of weight distributions in their neighborhood. Given the discussion above, this in turn leads to a change in the ORC pattern, as a group of edges with similar ORC values form. This indeed matches the observation in Figure 3, showing the emergence of more concentrated modes in the distribution of ORC following attention dynamics through a forward pass.

**Curvature and weight variance.** Recall that the ORC definition provided in Equation 15 expands the Wasserstein distance from Equation 2, which is the cost of optimal transport of a mass distribution from $\mathcal{N}_v$ to $\mathcal{N}_u$. Recognizing this, we can observe that, fixing any (unweighted) connectivity structure, the higher the imbalance in the neighborhoods, the larger their distributional distances, which leads to smaller curvature values considering the definition of ORC. Note that the connectivity in full attention is trivially fixed, since the graph is always fully connected. It follows that the curvature, in this case, depends on the (im)balance in weight distributions. Therefore, it is the *variance* of weight distribution that determines the curvature values in attention maps. This intuition informs the regularization proposed in Section 4.2 for training curvature-adjusted transformer models.

## D  IMPLEMENTATION DETAILS

### D.1  *L-ViT* CONFIGURATION

The *L-ViT* model utilizes a transformer architecture for image classification borrowed from ViT-PyTorch library (Wang, 2020). The input image is divided into 7x7 patches for the MNIST and Fashion-MNIST datasets, and 8x8 patches for the CIFAR10 dataset. These are projected into a 64-dimensional space within a single channel for the MNIST and Fashion-MNIST datasets, and 3 channels for the CIFAR10 dataset. An 8-layer deep transformer encoder with a single attention head per layer processes these patch embeddings. The MLP block within each layer has a hidden dimension of 128. Dropout with a rate of 0.1 is applied both to activation functions and embeddings during training to enhance model robustness and prevent overfitting.

## D.2 TRAINING AND EVALUATION DETAILS

We use a learning rate of $5 \cdot 10^{-4}$ for training *L-ViT* with a batch-size of 64 images, and a learning rate of $5 \cdot 10^{-6}$ for training BERT-Tiny with a batch-size of 16 sentences. The pre-trained model weights were downloaded from Huggingface Transformers (Wolf et al., 2020). For fine-tuning pre-trained vision models, we use a learning rate of $2.5 \cdot 10^{-4}$ with a batch-size of 64 images, and for pre-trained language models, these values are $2 \cdot 10^{-5}$ and 16.

The perturbations for experiments in Section 5 include flipping, blurring, and rotation of images with probabilities 0.2 and 0.4, and word swap in texts with probability 0.4. For the experiments in Section 4.2, we trained *L-ViT* for 20 epochs on MNIST and Fashion-MNIST images and 40 epochs on CIFAR10 images, and similar experiments on BERT-Tiny (in Appendix E.1) were performed with 20 epochs for all three datasets. For the *L-ViT* experiments in Appendix E.3, models were trained with an early stopping condition at accuracy of 97%, 87%, and 50%, over MNIST, Fashion-MNIST, and CIFAR10 validation sets, respectively. BERT-Tiny models mentioned in Appendix E.3 were trained with an early stopping condition at accuracy of 88%, 84%, and 79%, on the IMDb, Headlines, and SST-2 validation sets, respectively. In Appendix E.3, we perturb the test set images by applying elastic transformation with $\alpha = 75$ and random perspective transformation with a distortion scale of 0.5, with a probability of $p = 0.5$. Text test set sentences are perturbed via characters or word swap. These substitutions were chosen randomly, potentially resulting in nonsensical combinations. This process aimed to introduce noise and potentially disrupt the inherent coherence and structure of the sentences. Both character-level and word-level perturbations were applied independently with a fixed probability of $r = 0.2$ for each. Consistent subset sizes were employed for all experiments involving ORC computation over subsets. For both image and text datasets evaluated with *L-ViT* and BERT-Tiny, we used a fixed subset of 1,024 samples.

## D.3 OTHER IMPLEMENTATION DETAILS AND SYSTEM SPECIFICATION

All ORC computations in this study were conducted using the GraphRicciCurvature(Ni et al., 2019) Python library. All loss optimizations were performed via Adam (Kingma & Ba, 2014), and the training and testing implementations use the PyTorch library (Paszke et al., 2019), with the default data split from the Torchvision package (Marcel & Rodriguez, 2010). While visualizing distributions, outliers and zeros were removed as needed for clarity of the distributions in the figures. The system specification for the computations is provided in Table 2.

Table 2: System specifications for the computations.

| | |
|---|---|
| CPU | Intel(R) Xeon(R) CPU @ 2.40GHz |
| GPU | Nvidia A100 SXM4 40GB |
| OS | AlmaLinux 9.3 |
| Architecture | x86_64 |

Due to computational costs associated with the resource demands of graph ORC computations, evaluations of several experiments were conducted on randomly selected subsets of the MNIST, Fashion-MNIST, and CIFAR10 image datasets and the IMDb, Headlines, and SST-2 text datasets. Additionally, the pre-trained models were fine-tuned for up to 2 epochs on each dataset, which already yields over 90% accuracy.

# E ADDITIONAL EXPERIMENTS

## E.1 CURVATURE-ADJUSTED TRAINING OF LANGUAGE TRANSFORMERS

In Section 4.2 we propose a regularization method to adjust the curvature distribution of the attention map in transformers. This adjustment is motivated by our theoretical results in Section 4.1, and the proposed variance regularization follows the intuition described in Appendix C. The experiments in Section 4.2 further demonstrate the impact of this regularization, which, in addition to validating our theoretical results, provides a tool for training better-performing or more robust transformers. In this appendix section, we repeat these experiments for a language model.

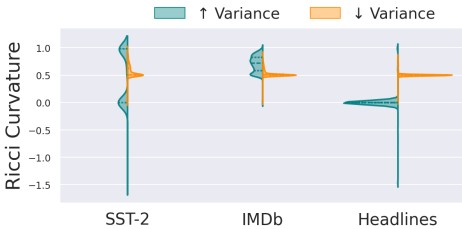

Figure 9: Final layer ORC distributions in BERT-Tiny trained via regularization methods increasing (↑) and decreasing (↓) variance, exhibiting a connection with the direction of variance regularization.

**Manipulating curvature distribution of language transformers.** The attention curvature distributions obtained through our proposed variance regularization applied to BERT-Tiny trained from initialized weights echo the same results as observed for vision models. Regularizing the loss to decrease the attention score variance discourages smaller curvature values as expected (Figure 9). This is observed for BERT-Tiny in two out of three datasets. Note that training language models from initialized weights requires substantially more computational resources than training L-ViT, and as a result, the performance does not reach the level of accuracy obtained by fine-tuning pre-trained language transformers. Furthermore, since the attention graph is substantially larger and curvature computation more expensive, we use a smaller sample size, leading to larger variance across data and trials.

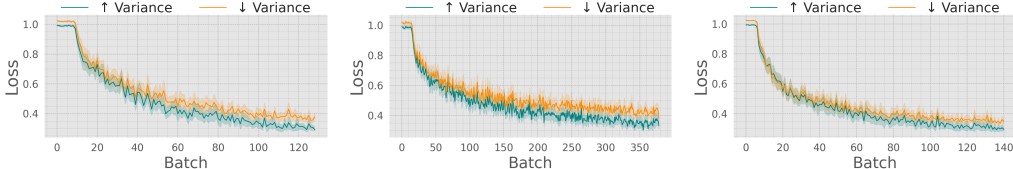

Figure 10: Loss minimization on BERT-Tiny with regularization methods increasing (↑) and decreasing (↓) variance for Headlines (left), SST-2 (middle), and IMDb (right). Higher variance can promote faster loss reduction and achieve convergence in fewer iterations.

**Loss minimization in variance-regularized training of language transformers.** We also observe larger loss variance across trials in language models compared to vision models, and due to computational constraints, analyzing and reducing this variance is beyond the scope of this project, especially considering strong empirical validations already observed using *L-ViT*. As a result, the loss dynamics for training BERT-Tiny with variance regularization, shown in Figure 10, does not (yet) yield statistically significant differences between increasing and decreasing variance. Having said that, the average over 30 trials on BERT-Tiny is consistent with what we observe for *L-ViT* in Section 4.2, though these average values fall within the 95% confidence interval of each other.

### E.2 FORGETFULNESS AND CURVATURE DISTRIBUTION

As a part of our empirical analysis of the robustness properties of transformers, in Section 5 we investigated the connection between ORC distribution and *forgetfulness* of pre-trained transformers. Recall that for this experiment, we first fine-tune each model on a primary dataset, and subsequently fine-tune it on a secondary dataset, monitoring how performance on test sets on both datasets change throughout the second round of fine-tuning. A fast decay in performance on the primary dataset implies a more *forgetful* model. As discussed in Section 5, our results indicate that DeiT and BERT-Tiny tend to forget learned patterns faster than ViT and ELECTRA-Small, signaling higher robustness of ViT and ELECTRA-Small to training on new distributions.

Meanwhile, as shown in Figure 6, the ORC distributions of attention maps in ViT and ELECTRA-Small lean further towards more positive ORC values. Together, these observations support our arguments regarding the positive correlation between the ORC of attention maps and the robustness

of transformers, as the more positive-leaning distribution belongs to the less forgetful model. The forgetfulness observations are demonstrated in Figure 7 for CIFAR10 and Headlines as the primary datasets, and MNIST and SST-2 as secondary datasets. We now include results on the other pairs of primary and secondary datasets. As Figure 11 demonstrates these results for vision transformers, the same trend as in Figure 7 is observed on most other pairs of datasets, further validating our results from this experiment. Figure 12 shows the results on other pairs for language transformers, where the same general trend is observed when forgetting occurs. It is worth noting that the forgetting phenomenon is not observed in some of our experiments on language models, which is likely due to semantic similarities between the tasks and datasets in hand.

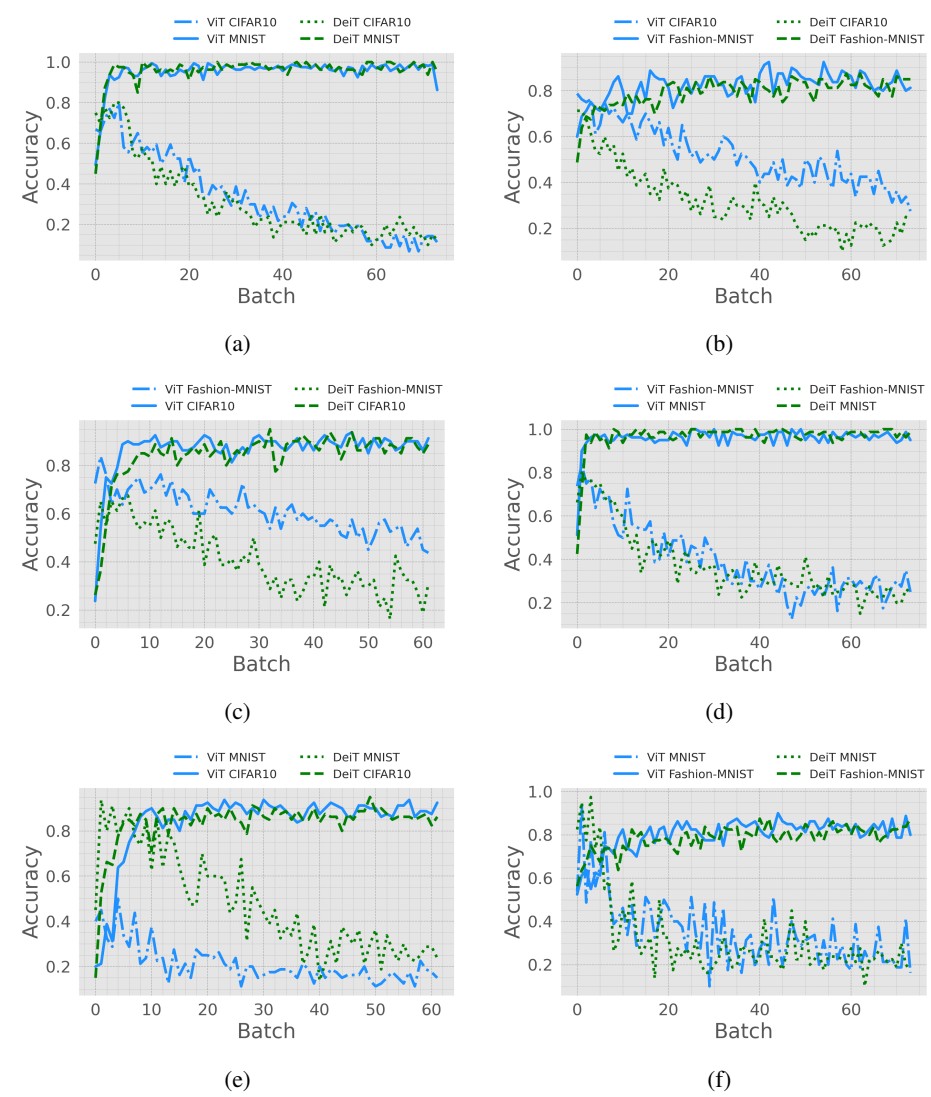

Figure 11: Test set accuracy of ViT (blue) and DeiT (green) models while fine-tuning on a secondary dataset after fine-tuning on a primary dataset. The primary and secondary datasets are: (a) CIFAR10 and MNIST; (b) CIFAR10 and Fashion-MNIST; (c) Fashion-MNIST and CIFAR10; (d) Fashion-MNIST and MNIST; (e) MNIST and CIFAR10; (f) MNIST and Fashion-MNIST. DeiT demonstrates a faster degradation in performance compared to ViT following subsequent fine-tuning.

### E.3 TRAINING ROBUST TRANSFORMERS

Given the established success of connection-dropping methods in training robust neural networks (Goodfellow et al., 2014; Jain et al., 2015; Srivastava et al., 2014; Yang et al., 2020), we also

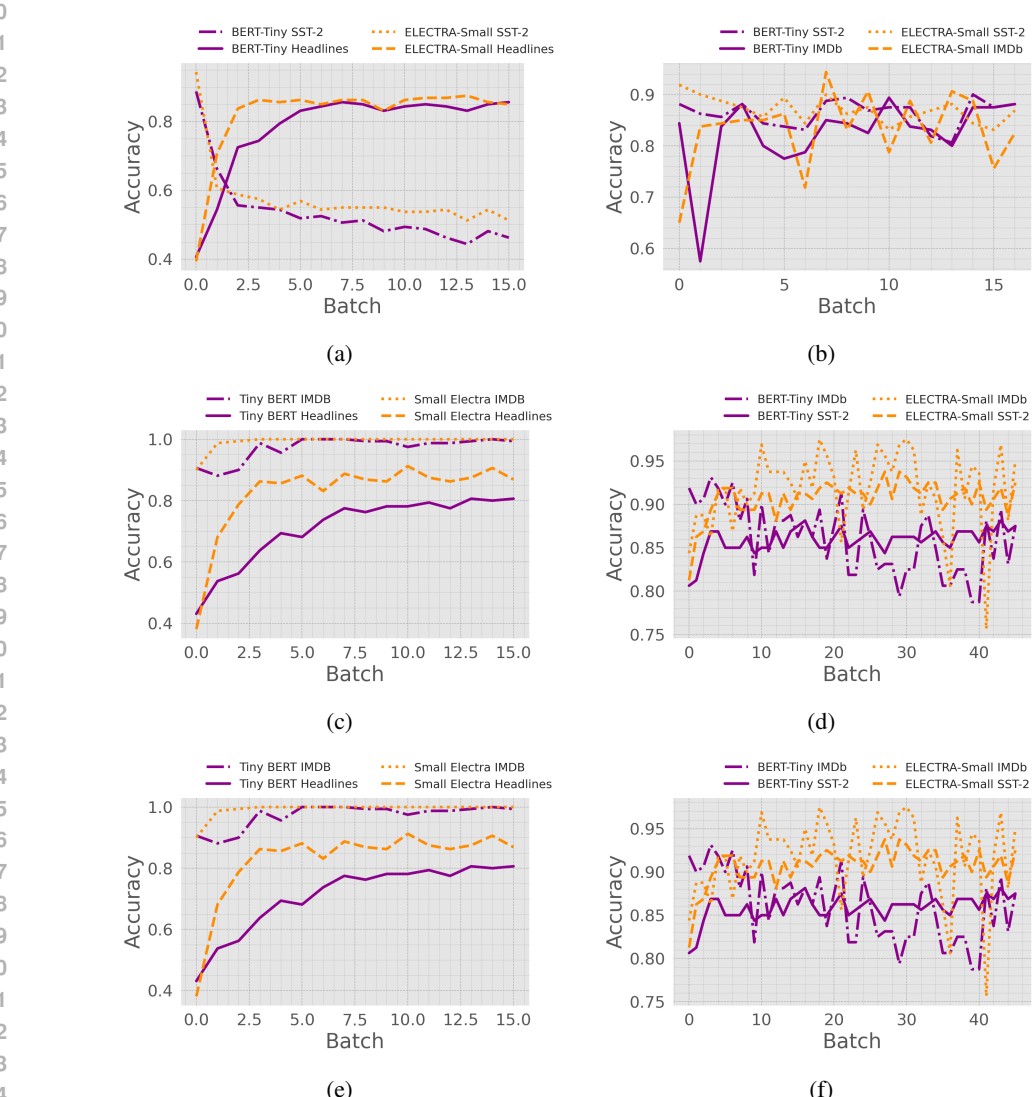

Figure 12: Test set accuracy of BERT-Tiny (purple) and ELECTRA-Small (orange) models while fine-tuning on a secondary dataset after fine-tuning on a primary dataset. The primary and secondary datasets are: (a) Headlines and SST-2; (b) Headlines and IMDb; (c) SST-2 and Headlines; (d) SST-2 and IMDb; (e) IMDb and Headlines; (f) IMDb and SST2. In cases where forgetting happens, BERT-Tiny demonstrates a faster degradation in performance compared to ELECTRA-Small following subsequent fine-tuning.

investigate the impact of random dropout of attention connections on the ORC distribution in *L-ViT* and BERT-Tiny. During training, a Bernoulli mask was implemented to stochastically remove edges across all attention maps at varying rates. In the testing phase, we assess the robustness of the trained models with unmanipulated attention, quantified by $\rho$ as defined in Equation 8, over dataset subsets (see Appendix D.2). Informed by prior research on connection masking techniques (Luo et al., 2021; Zehui et al., 2019), we posit that its application will lead to enhanced model robustness. While the number of active attention connections may decrease, we anticipate this effect will not significantly impair overall performance.

Observing the $\rho$ values at different Bernoulli masking rates on Figure 13, our experiments confirm that this robustness tends to be achieved. Noticeably, a trend toward increasing robustness is often observed when masking rates fall within the range of 0.2 to 0.5, indicating that masking at these rates promotes increased robustness. Furthermore, low performance degradation is observed even at

very high masking rates. This phenomenon can be attributed to two counteracting effects. Firstly, the masking process reduces the number of active edges, potentially leading to a decrease in overall model capacity. Secondly, the remaining connections become less redundant due to the sparsity introduced by masking. This may allow individual connections to exert a more significant influence on the model's robustness. Meanwhile, as we increase the masking rate, we see in Figure 14 that the negative values of ORC diminish. Furthermore, the ORC distributions of the final attention layer exhibit a convergence towards modal values across all examined datasets, which suggests a diminishing extremity within these distributions. This provides another piece of evidence for our claims on the connection between ORC in the attention map and the robustness of the transformer.

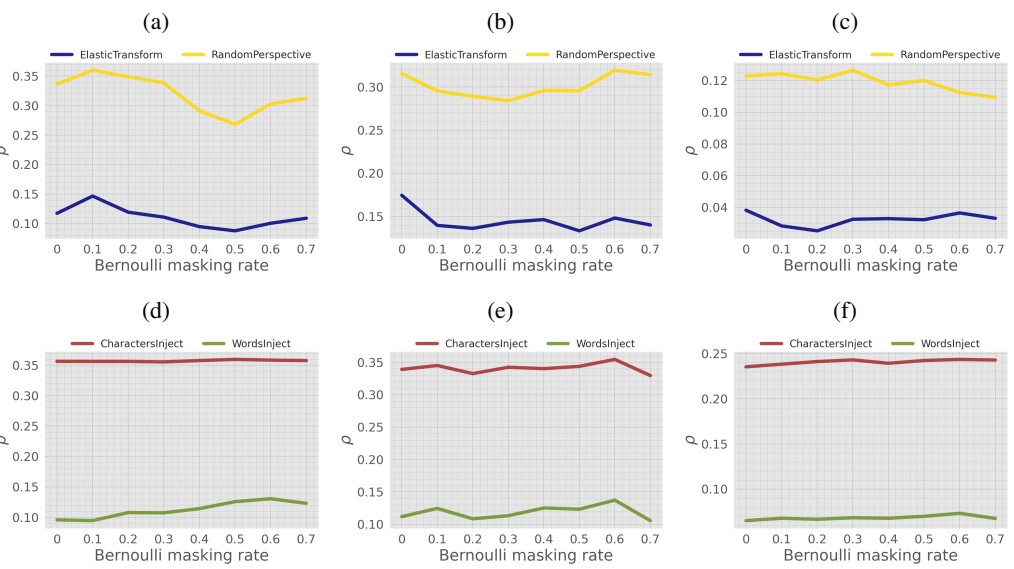

Figure 13: Perturbed test set accuracy rate degradation ($\rho$ as defined in Equation 8) of *L-ViT* (top row) and BERT-Tiny (bottom row) models trained with Bernoulli masking. Results are evaluated over random subsets of the: (a) MNIST; (b) Fashion-MNIST; (c) CIFAR10; (d) IMDb; (e) SST-2; and (f) Headlines datasets. Masking rates affect model robustness in both vision and language models.

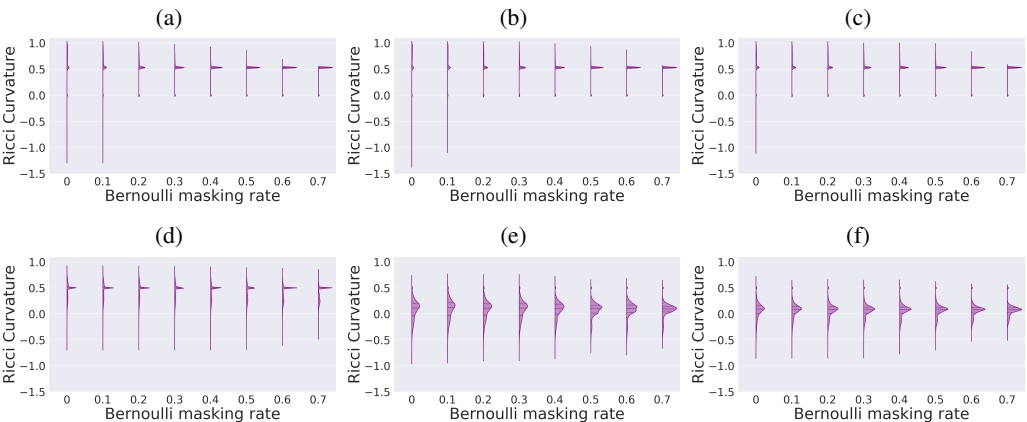

Figure 14: ORC distributions of the final layers of *L-ViT* (top row) and BERT-Tiny (bottom row) models trained with varying edge masking rates. Distributions are combined over random subsets of the: (a) MNIST; (b) Fashion-MNIST; (c) CIFAR10; (d) IMDb; (e) SST-2; and (f) Headlines datasets. Higher edge masking rates lead to curvature values convergence to a common modal value. Consequently, negative curvature values tend to diminish in both vision and language models.

# F RELATED WORK

In this work, we investigate the connections between the geometry of attention maps in transformers, their training, performance, and robustness properties. While related work in transformers, geometric deep learning, and robustness is referenced throughout the paper where relevant, we now provide additional discussion of the most related works in each area here.

**Transformers and attention mechanism.** Transformers (Vaswani et al., 2017) have revolutionized deep learning, achieving state-of-the-art performance across various domains (Brown et al., 2020; Devlin et al., 2018; Dosovitskiy et al., 2020). Recent works have begun to shed light on their inner workings, providing insights into their expressive power (Zaheer et al., 2020), generalization properties (Li et al., 2023), in-context learning (Von Oswald et al., 2023), learning dynamics (Abbe et al., 2024; Dovonon et al., 2024), and robustness characteristics (Bhojanapalli et al., 2021; Shao et al., 2021). However, many questions regarding the fundamental principles governing the behavior of transformers remain unanswered. Our work contributes to this growing body of research by investigating the connection between the geometry of attention maps and the properties of transformers.

**Graph Ricci curvature.** Ricci curvature captures the deviation of the local geometry from Euclidean space (Bauer et al., 2017). When extended to discrete structures like graphs, it has proven to be a powerful tool for various deep learning tasks (Di Giovanni et al., 2023; Liu et al., 2023; Southern et al., 2023; Topping et al., 2021). We employ the Ollivier-Ricci curvature (ORC) (Ollivier, 2009), an optimal transport formulation of Ricci curvature on graphs, to study the geometry of attention maps in transformers. To the best of our knowledge, this is the first work to explore this connection.

**Curvature and robustness.** Ricci curvature has been linked to the robustness of systems (Pouryahya et al., 2017), owing to its relationship with entropy. This connection is highlighted by a fundamental result from optimal transport theory (Lott & Villani, 2009), which provides bounds on entropy based on a lower bound of the Ricci curvature. The Fluctuation Theorem (Evans et al., 1993) further establishes a positive correlation between the robustness and entropy of a system, implying that robustness and curvature are positively correlated (Pouryahya et al., 2017). Leveraging this connection, we explore the implications of the ORC distribution for the robustness of vision and language transformers. In summary, our key contribution is uncovering insightful patterns in the curvature distribution of attention maps that evolve during training and through the layers of transformers, establishing a connection between the geometry of attention and the properties of these models. To the best of our knowledge, this is the first work to explore this connection, opening up new avenues for developing more robust and interpretable transformers.

