# OpenReview forum: "The Geometry of Attention: Ricci Curvature and Transformers Training and  Robustness"
_ICLR.cc/2025/Conference — ICLR 2025 Conference Withdrawn Submission_

### Official Review · Reviewer_1Qay · 2024-10-22

**Soundness:** 2
**Presentation:** 4
**Contribution:** 3
**Rating:** 5
**Confidence:** 4

**Summary:**

This paper addresses the relationship between the Olivier Ricci Curvature (ORC) of the attention graphs in transformers and their learning dynamics as well as their robustness to instance and distributional perturbations. In particular, the main hypothesis of the paper is more negatively distributed ORC values implies faster convergence speed of gradient descent in transformers, while more positively distributed ORC values leads to their more robustness. The authors have proposed a theoretical framework to support this hypothesis as well as a practical technique to incorporate it within training transformers. They also performed multiple experiments to empirically support their hypothesis.

**Strengths:**

1. This paper addresses a very interesting and important topic re: the interplay of transformers' trainability and their robustness and their relationship with the geometry of the attention maps. To my knowledge, this topic hasn't been explored explicitly in the literature which makes this work a valuable contribution in that regard.

2. The authors have also run extensive empirical studies to measure the relationship between transformers' robustness and training dynamics to geometric properties of the underlying attention maps quantified by ORC distribution. For robustness, the authors have covered various types of domain-specific perturbations and data augmentation techniques as well as generalization robustness, which is quite interesting and extensive, in my opinion. Furthermore, the link between the geometric structure of attention matrix and its training dynamics during gradient descent is very exciting.

3. The paper is generally very well written with a clear flow and precise statements.

**Weaknesses:**

Despite its interesting topic of study and the extensive empirical experiments, there are certain flaws in this work in terms the theoretical contribution of the paper which give me a pause to give a higher score to it:

1- First of all, it's not clear to me why Ricci Curvature has been chosen as the main cornerstone of impact on trainability and robustness in this work. There's nowhere in the proposed framework that ORC is directly being optimized nor injected into the training process. Also, the theoretical results clearly point to the smallest eigenvalues of normalized Laplacian which are directly linked to the notion of graph-cut and sparsity in the underlying graph. Sure enough, ORC is also strongly correlated with these fundamental quantities, but why not focusing on these fundamental notions instead of ORC? In particular, what would formulating the problem in terms of ORC give us specifically that we'd not be able to get using e.g. (weighted) graph cut, which is more general? From that view, it also seems the extensive previous work on sparse and low-rank attention is highly relevant to the current work, and yet there's no mention of them in the paper.

2- The main theoretical contribution of the paper might as well be true but the presented proof for it is indeed flawed. Please note that the assumption of having identical attention blocks in each layer does *not* imply that the actual attention matrix A would be the same in each layer, as A is dependent on the input of each layer which is different for each layer, even when using the same attention block. Therefore, in Eq. (5) & (9), the attention matrix A needs to be layer-dependent, which would break down the whole proof strategy. Possible ways of getting out of this are either to assume having the same attention matrix in each layer (which is completely unrealistic due to what we know of the forward pass in transformers) OR do the analysis for L=1 layer, but this also would break down the proof of Lemma 4.2. As a result, the current flawed proof would cast serious doubt on the main theoretical result of this paper unless the authors can provide another proof strategy.

3- As for the algorithmic contribution of the paper, it's not hard to see that given the standard L2-norm regularization on the network's weight, tuning the temperature parameter of the Softmax can also be used to modulate the variance of attention weights in an attention matrix. In particular, higher temperatures reduce the variance while lower temperatures increase it. So then the question is why not using the more natural SoftMax temperature instead of introducing ad-hoc regularization functions on the weight variance? I can see at least 3 main benefits of doing so over the proposed method in the paper: (A) we can have layer-specific adjustments, (B) instead of having a binary mode (variance increasing vs. variance decreasing), tuning the temperature parameter gives us a full continuous range of scenarios, and (C) we can actually learn the optimal temperature parameters for each layer in a completely data-centric fashion. For instance, for training a more robust model, we can use data perturbation/augmentation during training, and let the model decide what the temperature should be in each layer for that level of perturbation. At the very least, an empirical comparison between these two methods would be more persuasive.

4- As for the empirical evaluation, Table 1 is not really conclusive. I'd highly recommend using training loss instead of training accuracy (to measure training convergence) and use the test accuracy (and NOT the train-test accuracy difference) as the measure of generalizability which is the standard way of doing it. Also, please specify the values of hyper-parameters related to the regularization terms in your experiments and how you picked them.

Typos:

Line 176: positive-learning --> positive leaning

Line 716: positively --> negatively

**Questions:**

To recap my questions from the concerns raised in the previous section:

1- In your analysis and conclusions, what specific benefit ORC would give us compared to the more general notion of (weighted) graph cuts)? And how would that link your work to the existing extensive work on sparse and low-rank attention?

2- Does the main theoretical result of the paper sill hold given the flawed proof provided? If so, what's your proof strategy?

3- Why not using more natural SoftMax temperature instead of the proposed regularization terms, especially given its extra benefits? What particular benefit would we get from using the proposed regularization terms instead?

---

> ### Author Response · Authors · 2024-11-21
>
> We thank the reviewer for their careful review, their encouraging words, their helpful questions, and their constructive comments on the shortcomings. The following addresses their specific points.
>
> ---
> ---
> **Weaknesses:**
>
> 1. Since curvature is derived from attention values, we cannot directly manipulate it without modifying the attention scores. Given that we consider the attention scores as edge weights in a graph, manipulating curvature values by changing attention values in an isolated fashion would be a significant departure from our approach. Instead, we explore strategies to analyze the role of curvature on the properties of this graph (and consequently the transformer), and adjust these properties by indirectly modifying the curvature values, e.g., by penalizing attention variance. Analyzing attention maps as graphs does enable us to use a variety of tools and measures that characterize the structure and dynamics of graphs. While we do not suggest that curvature is the only useful one, it has certain properties of interest. We used some of these properties in this paper highlighting the utility of this measure, and others could be used in future research.
>
> - Curvature characterizes the local geometry of the graph. While related to bottleneckedness (hence to Cheeger constant, graph cut, etc.), curvature is more local. The alternatives mentioned do not capture the graph structure at the local level that curvature does.
>
> - Curvature is linked to three (related) notions of robustness on graphs: structural robustness, dynamic/system robustness, and distributional robustness.
>
> - Curvature is linked to fundamental concepts such as entropy, as we discuss; although it is not the only one.
>
>
> 2. We appreciate pointing this out. The assumption you refer to is in fact that $A$ is fixed across layers, not only that the weights are shared. This assumption is borrowed from Dovonon et al. That being said, we agree with your suggestion, we will reconsider this strong assumption, and will work on revising our proof strategy to relax it.
>
> 3. Thank you for this valuable suggestion. Variance regularization is a tool we used to modify the curvature distribution and investigate its impact on the behavior of transformers. We will further clarify this upon revision and resubmission. There are other tools for this, and we do not claim that variance is the most versatile tool, but it is sufficiently simple and effective for the purpose of our investigation. The softmax temperature and attention score variance are indeed related, and while variance regularization could also be layer-specific, the tunability and learnability of temperature are helpful advantages. Exploring how to modify the curvature distribution by adjusting the softmax temperature and other measures is an interesting direction that could strengthen and supplement our empirical observations. We will consider this direction in future work.
>
> 4. We will report training loss and will report the hyperparameters. For measuring generalizability, it is our understanding that for comparing two cases that do not reach the same level of training accuracy, test accuracy alone does not fully reflect generalizability.
>
> ---
> ---
> **Questions:**
>
> 1. Please see our response to weakness 1 above.
>
> 2. Please see our response to weakness 2 above.
>
> 3. Please see our response to weakness 3 above.
>
> ---
> ---
> We will make all the above points clearer in our future submission.

---

### Official Review · Reviewer_fWKP · 2024-10-31

**Soundness:** 4
**Presentation:** 4
**Contribution:** 3
**Rating:** 6
**Confidence:** 3

**Summary:**

This work focuses on the geometry of attention maps viewed as weighted graphs and analyzes the performance and robustness of transformers by considering the attention mechanism as a graph operator. This work establishes a link between Ricci curvature and the convergence behavior of gradient descent by theoretical analysis and highlights the impact of curvature on robustness. A trade-off between the performance and the robustness of the transformer model can be uncovered by the Ricci curvature. Hence, this is an interesting work.

**Strengths:**

Originality: The idea of using Ricci curvature to metricize the training dynamics and robustness of transformers is innovative and establishes a direct link between Ricci curvature and the convergence behavior of gradient descent in transformers.

Quality and Clarity: The quality of this is satisfactory and the analysis is thoughtful. The proposed method that manipulates the curvature distribution of attention maps through a variance-adjusting regularization term is sound and illuminates the trade-off between performance and robustness by adjusting the Ricci curvature.

Significance: This work provides a contribution to the transformer model, including studying and improving attention mechanisms.

**Weaknesses:**

It would be interesting to see if the proposed method outperforms other attention map distribution regularizations in the experiments.

**Questions:**

it is the same as Weaknesses.

---

> ### Author Response · Authors · 2024-11-21
>
> We thank the reviewer for their careful consideration of our work and for their encouraging remarks. The following addresses their specific weaknesses and questions.
>
> ---
> ---
> **Weaknesses and Questions:**
>
> This is a question that could be explored, and we find it intriguing to explore relevant metrics (e.g. entropy). The regularization method we discuss showcases an example of possible ways to modify the curvature distribution and how this could impact the behavior of transformers. We will clarify this and point to other possibilities upon revising our paper.

---

### Official Review · Reviewer_dNDf · 2024-11-01

**Soundness:** 1
**Presentation:** 2
**Contribution:** 2
**Rating:** 3
**Confidence:** 5

**Summary:**

The paper reinvests the idea of considering attention matrices as adjacency matrices of graphs to investigate potential connections between curvature and training dynamics in a layer that is claimed to resemble the one in transformers. A regularization scheme is introduced based on an observed correlation between the robustness of the network and the curvature. Some experiements are provided to support these claims.

**Strengths:**

- This paper’s aim which is to investigate the underlying graph structure explaining token's interactions is a compelling idea that deserves further exploration.

- Given the rich literature on curvature, geometry and graphs, this approach has the potential to uncover interesting phenomena by connecting these areas.

- Analyzing training dynamics from an overparametrized perspective is a promising approach and connects well with recent findings in optimization theory.

**Weaknesses:**

The main reason for rejecting this paper is that although it presents itself as a theoretical paper, it actually contains many imprecise, non rigorous and at times incorrect elements, below are some of the most problematic ones:

1. Some assumptions are overly restrictive, e.g. considering constant attention across layers (i.e. the attention cannot result from the standard key-query dot product involving the features from the previous layer) and bold claims are made throughout the paper to justify them, like in line 253 "Note that relaxing the above mentioned assumptions is rather straightforward, though distracting and not amenable to the here provided analysis". However, equation 6 immediately breaks without this assumption as computing the eigenvalues of a product of matrices is highly non trivial without further assumptions, despite the authors’claim, which is thus unsupported.

2. Additional issues with the paper’s assumptions include: (a) being relegated to the appendix, while the work is meant to be theoretical, (b) being labelled as «  standard » and constantly justified by the fact that they were used in Dovonan et al 2024, a paper that got rejected at ICLR 2024 precisely because of these limited assumptions (see discussion on openreview here https://openreview.net/forum?id=OCx7dp58H1), (c) being so restrictive that considering them reduces the analysis to considering a linear model in the input data, already analysed in which Liu et al., 2022 for (S)GD convergence.

3. Theorem 4.1 mis-cites the result from Liu et al. 2022--it holds only for the mean square error loss. In particular, the extension of such result to more general loss functions is highly non trivial and lead to multiple research papers (eg see Robin et al., 2022 and Scaman et al.,2022 where the authors generalize the result to cross entropy loss, among others). Additionally, in the experiments section, the authors use the cross entropy loss and dont seem to be fully aware of the aforementioned sublety (or at best a discussion is missing in the paper regarding the task under study).

4. The proof (Appendix A.2) of the main theorem (Lemma 4.2) is incorrect:
- Minor: line 762: "2L-3" should be "2L-2" line 773
- Major (this mistake invalidates the proof): line 774: if the laplacian is defined as L = I - A, then its largest eigenvalue is 1-the smallest eigenvalue of A, not 1-the largest eigenvalue of A, as incorrectly described in the paper. Given that the following arguments in the proof rely on a bound given in Bauer et al. 2011 that does not involve the largest eigenvalue of A, the proof likely requires substantial revision or correction.
- Major (invalidates the proof too): line 774: even if we were to assume the authors got a correct and meaningful bound between the smallest eigenvalue of A and the minimum ORC, the next argument in the proof does not hold either since it is based on differentiating an inequality.

5. The related work section in the appendix is not so well written and overly self-promotional. Each of its paragraph finishes with the same sentence "To the best of our knowledge, this is the first work to […]". In general, claimed contributions are overly emphasised on in this manuscript, reducing space for essential details and, at times, interfering with the quality of the writing.

6. Minor: the provided plots could be averaged over multiple simulations with visible some standard deviations (figures 3-4-6) to get more statistically convincing comparisons

Altogether, the study of attention layers' geometry shows potential, but significant revision and polish are needed, especially a substantial rework of the proof, before resubmission.

**Questions:**

- Taking a geometrical lens on the attention layer is an elegant thing on its own but could also potentially enable us to reveal and explain some phenomena. I would definitely encourage the authors to pursue in this direction.

- It may be helpful in the first place to restrict the analysis to a single-layer transformer, make the proof rigorous in this case, before aiming for more general results.

- In the definition of the graph ricci curvature, what led the authors to pick ORC as a good metric? And how do they define a distance metric on V, the set of vertices?

- The link between curvature and weight variance is unclear to me despite reading the appendix; could the authors clarify? Also, what makes this regularization term appealing?

---

> ### Author Response · Authors · 2024-11-21
>
> We sincerely thank the reviewer for their careful and constructive review. We value this feedback and will improve our paper based on the comments provided. We address the main concerns below and questions in the following comment under this thread, all of which will be incorporated and clarified in our future submission.
>
> ---
> ---
> **Weaknesses:**
>
> 1. We understand that our comments on relaxing the assumptions may have been unclear and miscommunicated in the writing. We did not intend to suggest that our analysis holds as is (or with minor adjustment) if the assumptions are relaxed, but rather that relaxing the assumption to a less restrictive version and redoing the analysis can lead to similar results, which is admittedly non-trivial. We agree with the reviewer that relaxing the assumption is non-trivial, and our statement could lead the reader to underestimate what it takes to revise the analysis under relaxed assumptions. We will revise this statement and, following your suggestion, we will revise our proof strategy in order to relax this assumption.
>
> 2. We acknowledge that our analysis is conducted in an idealized setting. In our resubmission, we will include further discussion on the assumptions being restrictive and the purpose our analysis serves despite this. It is our understanding that analysis in idealized settings can inspire clues for better understanding or improved heuristics which can be empirically tested and lead to insightful findings. This is the approach we take, which is not only limited to the work by Dovonon et al, but also other works on transformers. For instance, Von Oswald et al. 2023 [1] derive their results under a specific construction, which leads to highly influential and insightful findings. That being said, we do not intend to downplay the restrictiveness of our assumptions, and we will clarify this in our revision.
>
> 3. Thank you for bringing this to our attention. We will certainly include a discussion on this point and will also work on supplementing our empirical observations with experiments using the MSE loss.
>
> 4. We sincerely thank the reviewer for identifying these points, and will make sure to address them properly.
>
> - Regarding the first major point, we seemed to have made a mistake (possibly a typo) in the indexing of ${\lambda^L}$, incorrectly using ${\lambda^L}_1$ instead of ${\lambda^L}_n$. This mistake has affected the next step of the proof, where we used a lower bound on ${\lambda^L}_1$ instead of an upper bound on ${\lambda^L}_n$ ---both a result from Bauer et al. 2011. We will carefully revise and fix our analysis following this correction. We will also revise and supplement the experiments regarding this result to validate the implications of our revised analysis.
>
> - Regarding the second major point, we recognize that this needs either filling a gap in the proof or a clarification and revision of the presentation of our results to distinguish rigorous claims from educated guesses. From a computational perspective, the main intended outcome of this analysis was to optimize a bound as a surrogate for adjusting a target variable, where the bound is straightforward to compute but the target variable is not. For stating a rigorous statement, we agree with you; there is a missing step in the proof. This gap could be filled under a viable assumption, and we can supplement our experiments to test the practical applicability of this assumption. However, given the feedback provided on the set of assumptions for our analysis, we will instead consider limiting the theoretical results to the bounds and instead discuss our argument for the hypothesized implications that we empirically investigate. We will also supplement our experiments to strengthen this investigation.
>
>
> 5. We will revise the related work to better discuss the literature and clarify the gap we aim to fill.
>
>
> ---
> > *”Altogether, the study of attention layers' geometry shows potential…”*
>
> Once again, we thank the reviewer for their valuable and constructive comments and we will revise and make the improvements mentioned above before resubmission.
>
>
> ---
> ---
> We will clarify all the points above in our future resubmission.
>
>
> ---
> ---
> **References:**
>
> [1] Von Oswald, J., Niklasson, E., Randazzo, E., Sacramento, J., Mordvintsev, A., Zhmoginov, A., & Vladymyrov, M. (2023, July). Transformers learn in-context by gradient descent. In International Conference on Machine Learning (pp. 35151-35174). PMLR.

---

> > ### Author Response · Authors · 2024-11-21
> >
> > ---
> > ---
> > **Questions:**
> >
> > > *”It may be helpful in the first place to restrict the analysis to a single-layer transformer…”*
> >
> > As we mentioned above, we find this a very helpful suggestion, and we will consider revising our analysis and proof strategy so that we could relax the assumptions.
> >
> > > *”In the definition of the graph Ricci curvature, what led the authors to pick ORC as a good metric?”*
> >
> > While there are various notions of graph Ricci curvature, ORC has properties and previously-developed tools/results that are amenable to the analysis we provided, for instance, the connection with the spectrum of the Laplacian.
> >
> > > *”And how do they define a distance metric on V, the set of vertices?”*
> >
> > The geodesic distance is by default the weighted shortest path.
> >
> > > *”The link between curvature and weight variance is unclear to me despite reading the appendix; could the authors clarify?”*
> >
> > Perhaps this connection is more clear using a cutoff on the weights, as we intended to visualize in Figure 2. Consider normalizing the weights to a maximum edge weight of 1. The right-most graph in Figure 2 with 0 variance has positively curved edges, while the left-most graph has a higher variance, and meanwhile edges with a lower ORC compared to the other two graphs. Weight variance is only a tool we use to modify the ORC distribution though, and not the only possible tool.
> >
> > > *”Also, what makes this regularization term appealing?”*
> >
> > The regularization is not aimed to be a method on itself to improve transformers, but rather to showcase how modifying graph curvature distribution through variance may affect the performance, training, and robustness of transformers. We do not claim this to be the only or best tool; it is a simple and effective one. Developing more effective tools in light of the discussions provided in our paper is a promising direction for future research.
> >
> >
> > ---
> > ---
> > Again, we will clarify all the points above in our future resubmission.

---

> > > ### Comment · Reviewer_dNDf · 2024-11-24
> > >
> > > I sincerely thank the authors for their honest responses. I will however maintain my score and would encourage the authors to proceed to further susbtantial explorations before resubmitting.

---

> > > > ### Author Response · Authors · 2024-11-24
> > > >
> > > > Thank you for reviewing and responding to our response. As we mentioned, the correction for the mistake and the steps that follow are straightforward. We already intended to withdraw and carefully incorporate the correction and improvements for our future submission though, as you suggested. Thank you once again for helping us improve our work.

---

### Official Review · Reviewer_uqbc · 2024-11-02

**Soundness:** 2
**Presentation:** 2
**Contribution:** 3
**Rating:** 5
**Confidence:** 3

**Summary:**

This paper proposed a connection between gradient descent convergence and Ricci curvature in transformers and applied geometric tools to improve the performance of the attention mechanism.

**Strengths:**

1. This paper takes attention maps as weighted graphs and investigates the geometric properties.

2. Experiments are conducted thoroughly, making the empirical results more robust.

**Weaknesses:**

$\bullet$ It is unclear whether this paper is intended to be primarily theoretical or experimental. A significant portion is devoted to technical background, much of which either summarizes existing results from other works or relies on rough assumptions, without providing deeper theoretical insights. The subsequent introduction of a regularization method for training curvature-adjusted transformers and related experiments adds further ambiguity. If the goal is to present a theoretical paper, I suggest including more original theoretical contributions. Alternatively, if the focus is on proposing an algorithm, consider abbreviating the technical background and dedicating more space to the regularization method and experimental findings.

Minor points:
1. In the paragraph following Lemma 4.2, "... Laplacian Bauer et al. (2011) " $\to$ "... Laplacian (Bauer et al. 2011) ".
2. Provide a more detailed description of Figure 3 within the main text, rather than in the appendix, as it is difficult to interpret Figure 3 with only the information available in the main body of the paper.

**Questions:**

1. In equation 3, you mentioned that $\underline{k}$ is the lower bound of the Ricci curvature and that a large ORC results in high entropy. However, this relationship is not entirely clear to me, as a large ORC does not necessarily imply a higher lower bound. In fact, a large ORC could still be compatible with a low lower bound. If $\underline{k}$ served as an upper bound instead, this relationship would make more sense. Could you provide further explanation on how a large ORC correlates with high entropy under these conditions?

2. You mentioned, “A rigorous understanding of how transformers process their input remains elusive, especially through mathematical tools offering practical solutions.” However, there are many existing mathematical tools, such as Centered Kernel Alignment (CKA). What are the specific advantages of using Ricci curvature over other tools? CKA, for instance, provides more straightforward results and interpretability compared to Ricci curvature. Could you elaborate on the unique insights that Ricci curvature offers in this context?

---

> ### Author Response · Authors · 2024-11-21
>
> We are grateful to the reviewer for their careful consideration of our work. We would like to remark that a main contribution of this work is exploring the robustness of transformers (in addition to their performance) using the graph geometry of attention maps. The following addresses the specific points the reviewer has raised.
>
> ---
> ---
> **Weaknesses:**
>
> > *”It is unclear whether this paper ...”*
>
> We believe our paper falls in the category of works in deep learning research that provide theory-informed heuristics for improving understanding, intuition, inspiration, and methodological developments. We rely on results from more theoretical papers to provide understanding and point to promising venues for improvements in more applied directions. To this end, we bring together theoretical results from different areas to present our theoretical arguments for the utility of a computational tool, in particular graph curvature, which enables a better understanding and further improvements of transformers.
>
> > *”Provide a more detailed description of Figure 3...”
>
> We agree that the description of Figure 3 seems insufficient. We will elaborate and clarify.
>
> ---
> ---
> **Questions:**
>
> 1. We appreciate this question and will clarify this in our resubmission. Note that Eq. (3) applies locally, hence a more positive-leaning ORC distribution is expected to correspond to local neighborhoods with larger lower bounds. Furthermore, our experiments that modify curvature distribution also modify the minimum ORC.
>
> 2. In this paper we consider the attention map as a weighted graph. This point of view opens the door to a rich toolset of graph analysis, including graph geometry. A particular advantage of graph curvature is that it characterizes the local geometry of the attention map, while many other graph analysis tools are less sensitive to local geometry and directly capture more global topological features. Moreover, as we explain in the background, curvature is linked to entropy and robustness, which are fundamental concepts.
>
>
> ---
> ---
> We will make all the above points clearer in our future submission.

---

### Official Review · Reviewer_KJ9p · 2024-11-03

**Soundness:** 3
**Presentation:** 3
**Contribution:** 2
**Rating:** 5
**Confidence:** 5

**Summary:**

This paper explores the geometry of attention maps in transformer models. Specifically, the authors examine the relationship between Ricci curvature in attention, the convergence rate of gradient descent in transformers, and model robustness. They observe that lower curvature correlates with reduced robustness and faster convergence. Additionally, they demonstrate how the Ricci curvature distribution in attention graphs impacts transformer robustness. The authors propose a training method that enables curvature adjustment within the attention graph, enhancing transformer performance and generalizability. Experimental results support the theoretical findings and highlight the advantages of the proposed training approach.

**Strengths:**

1. The paper provides an interesting approach to studying attention mechanisms in transformers using Ollivier-Ricci curvature.

2. Theoretical results are discussed, and extensive empirical results are provided to justify the theoretical results.

3. The paper is well-written with illustrative figures.

**Weaknesses:**

1. The authors rely on strong assumptions to establish Lemma 4.2 and Theorem 3.1. Notably, Assumption A0 implies that the attention matrix A is symmetric, which is uncommon in practical settings. Assumption A1 is even more restrictive, as it requires A to be invertible. The remaining assumptions are also quite limiting. Furthermore, the authors assume that A does not depend on the input X; however, in practice, the attention matrix A is indeed a function of X, which invalidates the proofs presented in the paper.

2. Connections between curvature and robustness are not new and already established in [1] and [2]. Thus, this is not a contribution of the paper.

3. Experiments on larger datasets beyond CIFAR10, MNIST, and Fashion-MNIST are needed are needed for vision models.

**Questions:**

1. The proofs provided in the paper are not new and are mostly borrowed from other papers. Given that experiments are not the strength of the paper, this can be a minus point.

2. Empirical comparisons to the baseline methods seem to be missing.

**References**

[1] Lloyd Demetrius and Thomas Manke. Robustness and network evolution—an entropic principle. Physica A: Statistical Mechanics and its Applications, 346(3-4):682–696, 2005.

[2] Maryam Pouryahya, James Mathews, and Allen Tannenbaum. Comparing three notions of discrete Ricci curvature on biological networks. arXiv preprint arXiv:1712.02943, 2017.

**Details Of Ethics Concerns:**

I have no ethics concerns about the paper.

---

> ### Author Response · Authors · 2024-11-21
>
> We would like to thank the reviewer for their review of our work. Next, we address the reviewer’s specific comments.
>
> ---
> ---
> **Weaknesses:**
>
>
> 1.  As you point out and we acknowledge in the paper, the theoretical analysis is performed in an idealized setting, as it is often the case in the community. We do not intend to underestimate the restrictiveness of the assumptions, and we will further clarify this upon revision. Also as you mentioned, we provide extensive empirical results, which aim to assess the applicability of our theoretically educated arguments in practical scenarios. We will also clarify the following points:
>
> > *“Assumption A0 implies that the attention matrix A is symmetric”*
>
> While this implies that the similarities matrix inside the softmax is symmetric, this assumption alone does not necessarily imply that the attention matrix is symmetric, due to the value matrix, $V$.
>
> > *”A1 is even more restrictive, as it requires A to be invertible”*
>
> We agree that invertibility is a restrictive assumption. Meanwhile, it's worth noting Dovonon et al. [1] assume this invertibility and state that this tends to hold in their experiment, and our theoretical analysis partially builds upon their findings and results.
>
> > *”the authors assume that A does not depend on the input X”*
>
> Respectfully, this seems to be misunderstood. As defined in Section 2, $A$ depends on the input through $Q=X W_Q$ and $K = X W_K$.
>
> In summary, while we agree on the restrictiveness of the assumptions, our approach is to utilize theory-informed heuristics for developing a better understanding and pointing to directions for methodological improvements, aligning with a common practice in deep learning research. Our theory aims to inform and inspire, and our empirical investigation seeks to verify whether the trends our theory suggests within idealized settings are observed in practical settings.
>
> ---
> 2. The link between curvature and robustness is indeed established in prior works and this is not against our novelty but in favor/supportive of our work. We discuss this connection in the Background section, citing the references you have provided, and do not claim it to be our contribution. However, this paper’s main contribution is building upon and exploring this link in the context of transformers.
>
> ---
> 3. We considered 3 image and 3 text datasets on 2 vision and 2 language models to provide ample empirical evidence. We will consider supplementing the image datasets to improve this paper for future resubmission.
>
>
> ---
> ---
> **Questions:**
>
>
> 1. We indeed leverage tools and results from existing papers we cited in areas such as learning theory and graph geometry. The proof we provide (in Appendix A.2) is itself new though, combining these tools for our specific purpose.
>
> 2. This paper does not aim to introduce a new model, but rather to improve our understanding of properties of existing architectures. Hence, we did not find a comparison to other baselines informative for our purpose.
>
>
> ---
> ---
> We will clarify all the points above in our future resubmission.
>
>
> ---
> **References:**
>
> [1] Dovonon, G. J., Bronstein, M. M., & Kusner, M. J. (2024). Setting the record straight on transformer oversmoothing. arXiv preprint arXiv:2401.04301.

---

### Comment · Reviewer_dNDf · 2024-11-24

I appreciate the authors' honest responses to my concerns, which, understandably, could not be fully addressed during the rebuttal period (given the limited time, I did not expect the incorrect steps in the proof to be resolved). I hope the authors find our feedback constructive and use it to refine their work for a future version, ideally with a corrected proof.

At this stage, I strongly recommend rejecting the paper due to the significant inaccuracies it contains (detailed in my review). As reviewers, it is our responsibility to maintain a high standard for accepted papers at ICLR, and accepting this submission would compromise that standard.

---

> ### Author Response · Authors · 2024-11-24
>
> Thank you for reviewing our response, and thank you again for your constructive feedback. As we explained in our response, we noticed the error following your feedback, and there is a clear path to fixing the error, which we will follow. Also, as we mentioned in our responses, we plan to do this carefully and resubmit in the future with the correction and improvements discussed. To clarify, by that we meant for another venue; we had already planned to withdraw. We did not withdraw right away because we wanted to have the opportunity to openly thank the reviewers for their superb feedback and inform them that all their comments are addressable but we needed more time. Appreciating the rebuttal process and the chance to have a discussion, we decided to respond and wait to hear more feedback on our response, but that was not with the intention to get this version of the paper published.
>
> In conclusion, as we had intended, we will withdraw, correct the error and incorporate other improvements discussed and will resubmit elsewhere.

---

### Author Response · Authors · 2024-11-24

We sincerely thank all reviewers and the AC for helping us improve our paper. We are encouraged by the recognition of our work’s strengths and grateful for the insightful comments and questions addressing the shortcomings and areas for improvement. According to all reviewers, we present a novel investigation pointing to an interesting and promising direction for analyzing and improving transformer models through the curvature and graph geometry of the attention map.

The reviewers provided careful and constructive feedback, which helped us identify the shortcomings and improve the paper for future submission. We take responsibility for these shortcomings, will incorporate corrections and clarifications, improve the presentation, and supplement the paper with further discussions to enhance the clarity of our contribution. For now, we have updated our submission with minimal changes to fix the error. While the correction and the improvements discussed during the rebuttal are clear, we plan to incorporate them carefully in an improved version for our future submission.

Please find our response to each individual reviewer under the reviewers' comments.

---

### Note · Authors · 2024-11-24

**Comment:**

We thank the reviewers for their time and valuable feedback on our submission. We appreciate the recognition of our work's strengths and the questions and comments regarding its shortcomings.

After careful consideration of the feedback received, we acknowledge that we need to improve the paper, make some clarifications and corrections, and enhance the presentation. We take full responsibility for the shortcomings, and have decided to withdraw our submission at this time. We will improve and resubmit a paper that resolves these shortcomings and more effectively reflects the value and implications of our research.

Thank you again for your comments.

**Withdrawal Confirmation:**

I have read and agree with the venue's withdrawal policy on behalf of myself and my co-authors.